# Assigning mitochondrial localization of dual localized proteins using a yeast Bi-Genomic Mitochondrial-Split-GFP

**Gaétan Bader[1‡§], Ludovic Enkler[1‡§], Yuhei Araiso[1†#], Marine Hemmerle[1†], Krystyna Binko[2], Emilia Baranowska[2], Johan-Owen De Craene[1¶], Julie Ruer-Laventie[3], Jean Pieters[3], Déborah Tribouillard-Tanvier[4**], Bruno Senger[1], Jean-Paul di Rago[4], Sylvie Friant[1], Roza Kucharczyk[2*], Hubert Dominique Becker[1*]**

[1]Université de Strasbourg, CNRS UMR7156, Génétique Moléculaire, Génomique, Microbiologie, Strasbourg, France; [2]Institute of Biochemistry and Biophysics, Polish Academy of Sciences, Warsaw, Poland; [3]Biozentrum, University of Basel, Basel, Switzerland; [4]Institut de Biochimie et Génétique Cellulaires, CNRS UMR5095, Université de Bordeaux, Bordeaux, France

**\*For correspondence:**
roza@ibb.waw.pl (RK);
h.becker@unistra.fr (HDB)

[†]These authors contributed equally to this work
[‡]These authors also contributed equally to this work
[**]Research Associate from INSERM

**Present address:** [§]Biozentrum, University of Basel, Basel, Switzerland; [#]Department of Clinical Laboratory Science, Division of Health Sciences, Graduate school of Medical Science, Kanazawa University, Kanazawa, Japan; [¶]EA 2106 Biomolécules et Biotechnologies Végétales, Université de Tours, Tours, France

**Competing interests:** The authors declare that no competing interests exist.

**Abstract** A single nuclear gene can be translated into a dual localized protein that distributes between the cytosol and mitochondria. Accumulating evidences show that mitoproteomes contain lots of these dual localized proteins termed echoforms. Unraveling the existence of mitochondrial echoforms using current GFP (Green Fluorescent Protein) fusion microscopy approaches is extremely difficult because the GFP signal of the cytosolic echoform will almost inevitably mask that of the mitochondrial echoform. We therefore engineered a yeast strain expressing a new type of Split-GFP that we termed Bi-Genomic Mitochondrial-Split-GFP (BiG Mito-Split-GFP). Because one moiety of the GFP is translated from the mitochondrial machinery while the other is fused to the nuclear-encoded protein of interest translated in the cytosol, the self-reassembly of this Bi-Genomic-encoded Split-GFP is confined to mitochondria. We could authenticate the mitochondrial importability of any protein or echoform from yeast, but also from other organisms such as the human Argonaute 2 mitochondrial echoform.

## Introduction

Mitochondria provide aerobic eukaryotes with adenosine triphosphate (ATP), which involves carbohydrates and fatty acid oxidation (*Saraste, 1999*), as well as numerous other vital functions like lipid and sterol synthesis (*Horvath and Daum, 2013*) and formation of iron-sulfur cluster (*Lill et al., 2012*). Mitochondria possess their own genome, remnant of an ancestral prokaryotic genome (*Gray, 2017*; *Margulis, 1975*) that has been considerably reduced in size due to a massive transfer of genes during eukaryotic evolution (*Thorsness and Weber, 1996*). As a result, most of the proteins required for mitochondrial structure and functions are expressed from the nuclear genome (>99%) and synthetized as precursors targeted to the mitochondria by mitochondrial targeting signals (MTS), that in some case are cleaved upon import (*Chacinska et al., 2009*). In the yeast *S. cerevisiae*, about a third of the mitochondrial proteins (mitoproteome) have been suggested to be dual localized (*Ben-Menachem et al., 2011*; *Dinur-Mills et al., 2008*; *Kisslov et al., 2014*), and have been named echoforms (or echoproteins) to accentuate the fact that two identical or nearly identical forms of a protein, can reside in the mitochondria and another compartment (*Ben-Menachem and Pines, 2017*). Due to these two coexisting forms and the difficulty to obtain pure mitochondria,

determination of a complete mitoproteome remains challenging and gave rise to conflicting results (*Kumar et al., 2002*; *Morgenstern et al., 2017*; *Reinders et al., 2006*; *Sickmann et al., 2003*).

Among all possible methods used to identify the subcellular destination of a protein, engineering green fluorescent protein (GFP) fusions has the major advantage that these fusions can be visualized in living cells using epifluorescence microscopy. This method is suitable to discriminate the cytosolic and mitochondrial pools of dual localized proteins when the cytosolic fraction has a lower concentration than the mitochondrial one (*Weill et al., 2018*). However, when the cytosolic echoform is more abundant than the mitochondrial one, this will inevitably eclipse the mitochondrial fluorescence signal. To bypass this drawback, we designed a yeast strain containing a new type of Split-GFP system termed Bi-Genomic Mitochondrial-Split-GFP (BiG Mito-Split-GFP) because one moiety of the GFP is encoded by the mitochondrial genome, while the other one is fused to the nuclear-encoded protein to be tested. By doing so, both Split-GFP fragments are translated in separate compartments and only mitochondrial proteins or echoforms of dual localized proteins trigger GFP reconstitution and can be visualized by fluorescence microscopy of living cells.

We herein first validated this system with proteins exclusively localized in the mitochondria and with the dual localized glutamyl-tRNA synthetase (cERS) that resides and functions in both the cytosol and mitochondria as we have shown previously (*Frechin et al., 2009*; *Frechin et al., 2014*). We next applied our Split-GFP strategy to the near-complete set of all known yeast cytosolic aminoacyl-tRNA synthetases. Interestingly, we discovered that two of them, cytosolic phenylalanyl-tRNA synthetase 2 (cFRS2) and cytosolic histidinyl-tRNA synthetase have a dual localization. We also confirmed the recently reported dual cellular location of cytosolic cysteinyl-tRNA synthetase (cCRS) (*Nishimura et al., 2019*). We further demonstrate that our yeast BiG Mito-Split-GFP strain can be used to better define non-conventional mitochondrial targeting sequences and to probe the mitochondrial importability of proteins from other eukaryotic species (human, mouse and plants). For instance, we show that the mammalian Argonaute 2 protein heterologously expressed in yeast localizes inside mitochondria.

## Results

### Construction of the BiG Mito-Split-GFP strain encoding the GFP$_{\beta1-10}$ fragment in the mitochondrial genome

We used the scaffold of the self-assembling Superfolder Split-GFP fragments designed by Cabantous and coworkers (*Cabantous et al., 2005b*; *Pédelacq et al., 2006*), where the 11 beta strands forming active Superfolder GFP are separated in a fragment encompassing the 10 first beta strands (GFP$_{\beta1-10}$) and a smaller one consisting of the remaining beta strand (GFP$_{\beta11}$). Seven amino acid (aa) residues of GFP$_{\beta1-10}$ and three of GFP$_{\beta11}$ were replaced in order to increase the stability and the self-assembly of both fragments (*Figure 1—figure supplement 1*). To increase the fluorescent signal and facilitate observation of low-abundant proteins, we concatenated and fused three β11 strands (GFP$_{\beta11\text{-chaplet; }\beta11ch}$) linked by GTGGGSGGGSTS spacers (see Materials and methods for DNA sequence, *Figure 1—figure supplement 1*, as in *Kamiyama et al., 2016*; *Figure 1A*).

Our objective was to integrate the gene encoding the GFP$_{\beta1-10}$ fragment into the mtDNA so that it will only be translated inside the mitochondrial matrix, while the GFP$_{\beta11ch}$ fragment is fused to the nuclear-encoded protein of interest and thus translated by cytosolic ribosomes (*Figure 1A*). To achieve this, we constructed a strain (RKY112) in which the coding sequence of the *ATP6* gene has been replaced by *ARG8m* (*atp6::ARG8m*), and where *ATP6* is integrated at the mitochondrial *COX2* locus under the control of the 5' and 3' UTRs of *COX2* gene (*Supplementary file 1*; *Table 1*; *Figure 1—figure supplement 2A–C*; see Materials and methods section for details). The RKY112 strain grew well on respiratory carbon source as wild type yeast (MR6) (*Figure 1B*), produced ATP effectively (*Figure 1C*), and expressed normally Atp6 and all the other mitochondria-encoded proteins (*Figure 1D*). We next integrated at the *atp6::ARG8m* locus of RKY112 strain mtDNA, the sequence encoding GFP$_{\beta1-10}$ (*Figure 1A*; *Figure 1—figure supplement 2*). To this end, we first introduced into the ρ$^0$ mitochondria (*i.e.* totally lacking mtDNA) of DFS160 strain, a plasmid carrying the *GFP$_{\beta1-10}$* sequence flanked by 5' and 3' UTR sequences of the native *ATP6* locus (pRK67, see Materials and methods for DNA sequence), yielding the RKY172 strain (bearing a non-functional synthetic ρ$^{-S}$ mtDNA, *Figure 1—figure supplement 2C*). This strain was crossed to RKY112 to enable

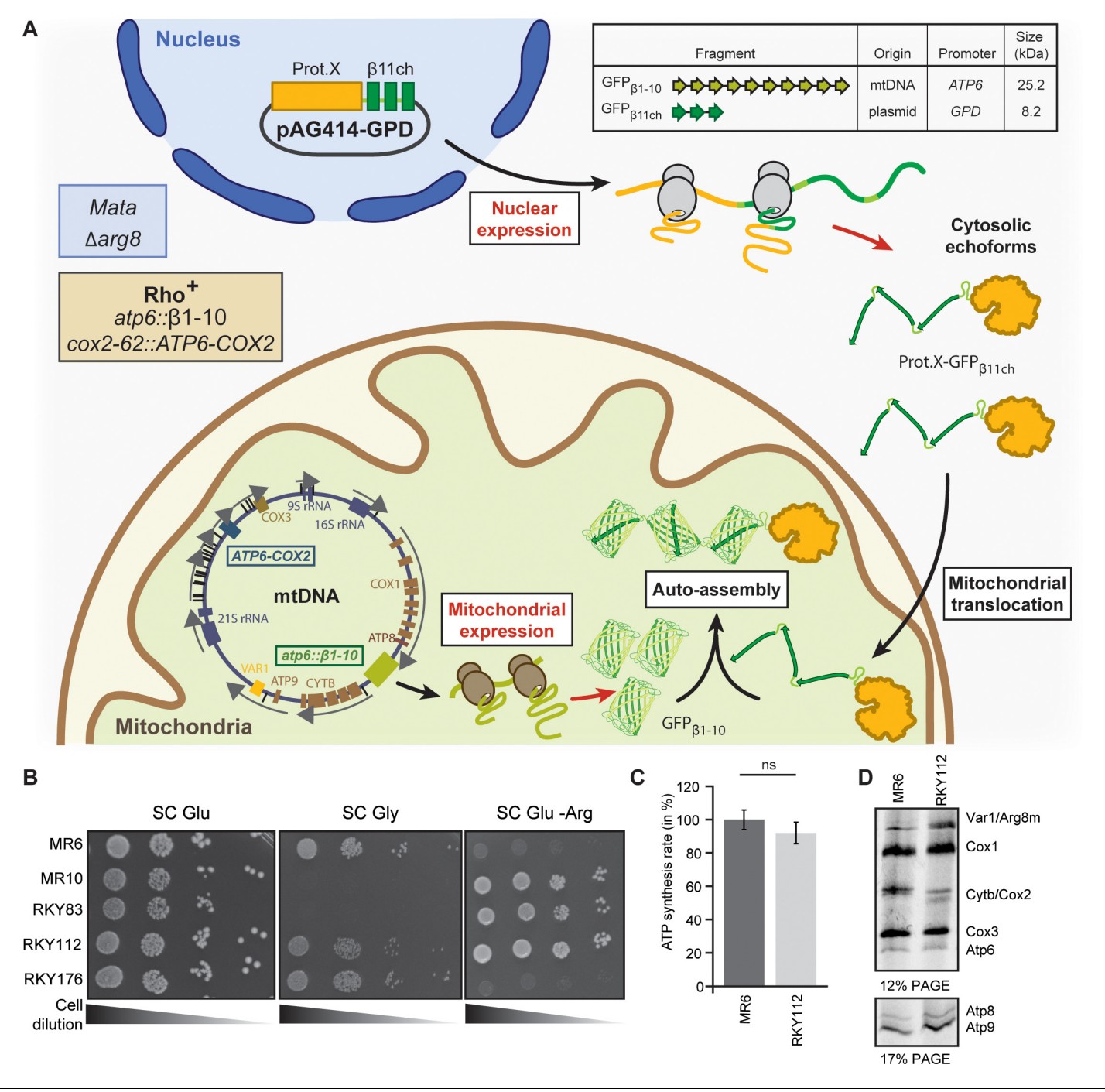

**Figure 1.** Engineering of the BiG Mito-Split-GFP system in *S. cerevisiae*. (**A**) Principle of the Split-GFP system. When present in the same subcellular compartment, two fragments of GFP namely GFP$_{β1-10}$ and GFP$_{β11ch}$ can auto-assemble to form a fluorescent BiG Mito-Split-GFP chaplet (three reconstituted GFPs). *GFP$_{β1-10}$* sequence encoding the first ten beta strands of GFP has been integrated into the mitochondrial genome under the control of the *ATP6* promoter. GFP$_{β11ch}$ consists of a tandemly fused form of the eleventh beta strand of GFP and is expressed from a plasmid under the control of a strong GPD promoter (pGPD). The molecular weight of the tag is indicated. (**B**) Growth assay on permissive SC Glu plates, respiratory plates (SC Gly), and restrictive media lacking arginine (SC Glu -Arg) of the different strains used in the study (N = 2). All generated strains are derivative from MR6. (**C**) ATP synthesis rates of the MR6 and RKY112 strains presented as the percent of the wild type control strain (N = 2). P-value was 0.7456 (not significant). 95% confidence interval was −273.4 to 229.9, R squared = 0.064 (**D**) Mitochondrial translation products in the MR6 and RKY112 strains (N = 2). Cells were grown in rich galactose medium. Pulse-chase of radiolabeled [$^{35}$S]methionine + [$^{35}$S]cysteine was performed by a 20 min incubation in the presence of cycloheximide. Total cellular extracts were separated by SDS PAGE in two different polyacrylamide gels prepared with a 30:0.8 ratio

*Figure 1 continued on next page*

*Figure 1 continued*

of acrylamide and bis-acrylamide. Upper gel: 12% polyacrylamide gel containing 4 M urea and 25% glycerol. Lower gel: 17.5% polyacrylamide gel. Gels were dried and exposed to X-ray film. The representative gels are shown.

The online version of this article includes the following source data and figure supplement(s) for figure 1:

**Source data 1.** Respiratory competency and translation of mtDNA-encoded respiratory subunits of the strains used in this study.

**Source data 2.** Statistics of the comparison of ATP synthesis rates between RKY112 and MR6 strains (related to *Figure 1C*).

**Figure supplement 1.** Optimized sequence and secondary structure of the $GFP_{\beta1-10}$ and $GFP_{\beta11ch}$ that were used in this study (related to *Figure 1*).

**Figure supplement 2.** Engineering of the strains and verification of the correct integration of *ATP6* under the control of *COX2* gene UTRs or $GFP_{\beta1-10}$ under the control of *ATP6* gene UTRs (related to *Figure 1*).

replacement of *ARG8m* with $GFP_{\beta1-10}$. The desired recombinant clones, called RKY176, were identified by virtue of their incapacity to grow in media lacking arginine due to the loss of *ARG8m* and their capacity to grow in respiratory media (*Figure 1B*). Integration of $GFP_{\beta1-10}$ in mtDNA was confirmed by PCR (*Figure 1—figure supplement 2E*, *Supplementary file 2*) and Western blot with anti-GFP antibodies (*Figure 2C*). Finally, the BiG Mito-Split-GFP strain (*Table 1*) was obtained by restoring the nuclear *ADE2* locus in order to eliminate interfering fluorescence emission of the vacuole due to accumulation of a pink adenine precursor (*Fisher, 1969*; *Kim et al., 2002*).

## The BiG Mito-Split-GFP system restricts fluorescence emission to mitochondrially-localized proteins

The BiG Mito-Split-GFP system was first tested with Pam16 which localizes in the matrix at the periphery of the mitochondrial inner membrane and Atp4, an integral membrane protein with domains exposed to the matrix (*Kozany et al., 2004*; *Velours et al., 1988*; *Figure 2A*). The BiG Mito-Split-GFP host strain was transformed with centromeric plasmids expressing either $Pam16_{\beta11ch}$

**Table 1.** Genotypes of yeast strains used or generated for this study.

| Strain | Nuclear genotype | mtDNA | Source |
|---|---|---|---|
| MR6 | *MATa ade2-1 his3-11,15 trp1-1 leu2-3,112 ura3-1 CAN1 arg8::HIS3* | $\rho^+$ | *Rak et al., 2007* |
| DFS160 | *MATα leu2Δ ura3-52 ade2-101 arg8::URA3 kar1-1* | $\rho^o$ | *Steele et al., 1996* |
| NB40-3C | *MATa lys2 leu2-3,112 ura3-52 his3ΔHindIII arg8::hisG* | $\rho^+$ cox2-62 | *Steele et al., 1996* |
| MR10 | *MATa ade2-1 his3-11,15 trp1-1 leu2-3,112 ura3-1 CAN1 arg8::hisG* | $\rho^+$ atp6::ARG8m | *Rak et al., 2007* |
| SDC30 | *MATα leu2Δ ura3-52 ade2-101 arg8::URA3 kar1-1* | $\rho^-$COX2 ATP6 | *Rak et al., 2007* |
| YTMT2 | *MATα leu2Δ ura3-52 ade2-101 arg8::URA3 kar1-1* | $\rho^+$cox2-62 | This study |
| RKY83 | *MATa ade2-1 his3-11,15 trp1-1 leu2-3,112 ura3-1 arg8::HIS3* | $\rho^+$cox2-62 atp6::ARG8m | This study |
| RKY89 | *MATα leu2Δ ura3-52 ade2-101 arg8::URA3 kar1-1* | $\rho^-S5'UTR_{COX2}$ ATP6 3'UTR_{COX2} COX2 | This study |
| RKY112 | *MATa ade2-1 his3-11,15 trp1-1 leu2-3,112 ura3-1 arg8::HIS3* | $\rho^+$ atp6::ARG8m 5'UTR_{COX2} ATP6 3'UTR_{COX2} | This study |
| RKY172 | *MATα leu2Δ ura3-52 ade2-101 arg8::URA3 kar1-1* | $\rho^{-S}$ atp6::$GFP_{\beta1-10}$ COX2 | This study |
| RKY176 | *MATa ade2-1 his3-11,15 trp1-1 leu2-3,112 ura3-1 CAN1 arg8::HIS3* | $\rho^+$atp6::$GFP_{\beta1-10}$ 5'UTR_{COX2} ATP6 3'UTR_{COX2} | This study |
| BiG Mito- Split-GFP | *MATa his3-11,15 trp1-1 leu2-3,112 ura3-1 CAN1 arg8::HIS3* | $\rho^+$atp6::$GFP_{\beta1-10}$ 5'UTR_{COX2} ATP6 3'UTR_{COX2} | This study |
| BiG Mito- Split-GFP+PAM16$_{\beta11ch}$ | *MATa his3-11,15 trp1-1-1::PAM16$_{\beta11ch}$ leu2-3,112 ura3-1 CAN1 arg8::HIS3* | $\rho^+$atp6::$GFP_{\beta1-10}$ 5'UTR_{COX2} ATP6 3'UTR_{COX2} | This study |
| BiG Mito- Split-GFP+PGK1$_{\beta11ch}$ | *MATa his3-11,15 trp1-1::PGK1$_{\beta11ch}$ leu2-3,112 ura3-1 CAN1 arg8::HIS3* | $\rho^+$atp6::$GFP_{\beta1-10}$ 5'UTR_{COX2} ATP6 3'UTR_{COX2} | This study |
| BiG Mito- Split-GFP+GUS1$_{\beta11ch}$ | *MATa his3-11,15 trp1-1:: GUS1$_{\beta11ch}$ leu2-3,112 ura3-1 CAN1 arg8::HIS3* | $\rho^+$atp6::$GFP_{\beta1-10}$ 5'UTR_{COX2} ATP6 3'UTR_{COX2} | This study |
| BY 4742 | *MATα his3Δ1 leu2Δ0 lys2Δ0 ura3Δ0* | $\rho^+$ | *Winston et al., 1995* |

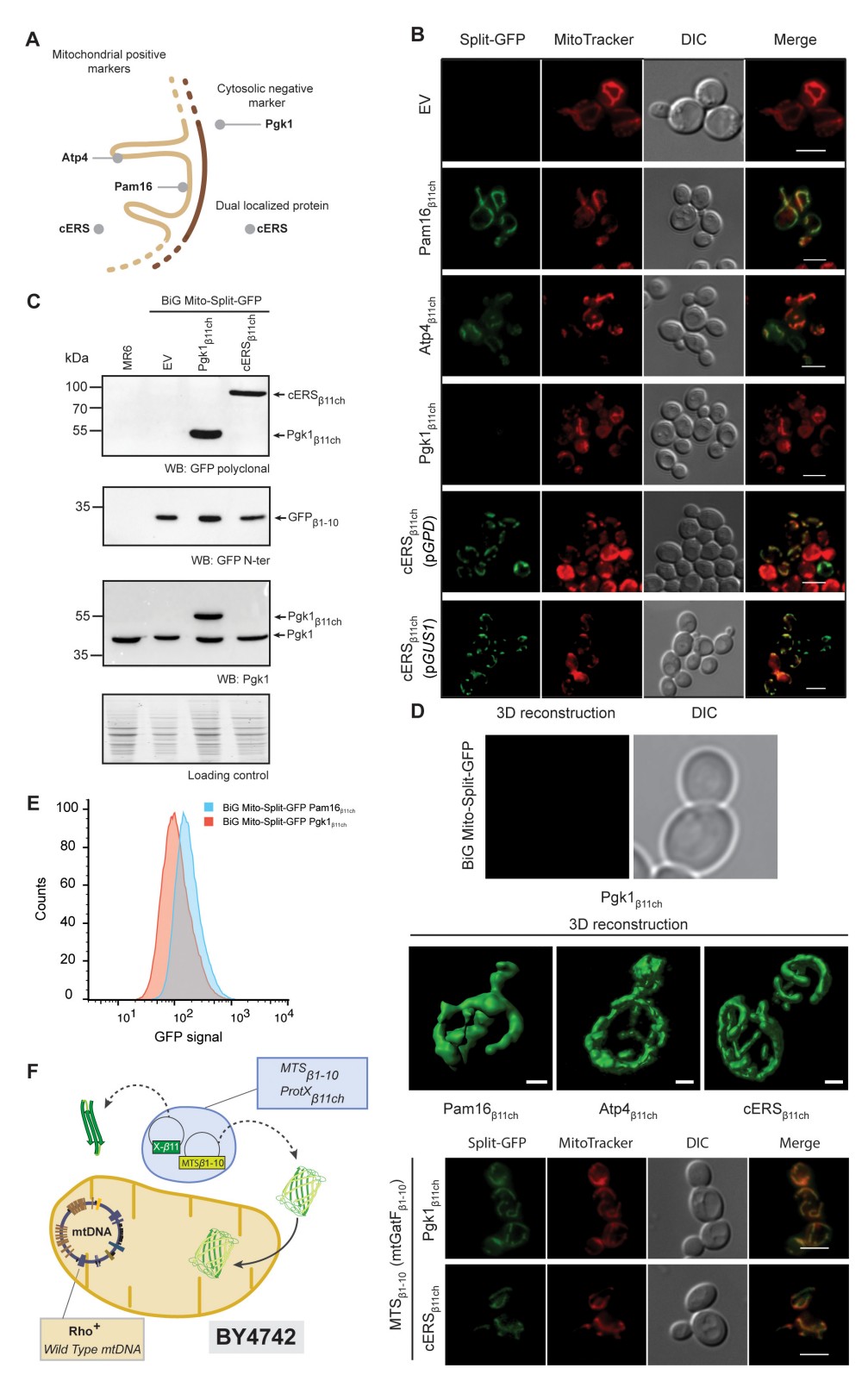

**Figure 2.** The reconstitution and fluorescence emission of the BiG Mito-Split-GFP is confined to mitochondria and exclusively generated by mitochondrial proteins. (**A**) Schematic of the spatial localization of proteins used as positive mitochondrial control proteins (Atp4, Pam16), negative cytosolic control protein (Pgk1) and as dual localized protein (cERS) in *S. cerevisiae*. (**B**) Empty pAG414pGPD$_{β11ch}$ vector (EV) or pAG414pGPD$_{β11ch}$ vectors expressing each of the four GFP$_{β11ch}$-tagged proteins used as markers in our study were transformed into the BiG Mito-Split-GFP strain.

*Figure 2 continued on next page*

**Figure 2 continued**

cERS$_{\beta11ch}$ was either expressed under the dependence of the GPD (pGPD) or its own promoter (pGUS1) from a centromeric plasmid. GFP reconstitution upon mitochondrial import was followed by epifluorescence microscopy (N = 3). (**C**) Immunodetection of the GFP$_{\beta1-10}$, cERS$_{\beta11ch}$ and Pgk1$_{\beta11ch}$ fusion protein in whole cell extract from the transformed BiG Mito-Split-GFP strain using anti-GFP and -Pgk1 antibodies, confirming expression of Pgk1$_{\beta11ch}$. Loading control: stain-free. The representative gels are shown. (**D**) The strains described in the legend of panel (**B**) were used for three-dimensional reconstitution of yeast mitochondrial network (N = 1). Z-Stack images from Pam16$_{\beta11ch}$, Atp4$_{\beta11ch}$, cERS$_{\beta11ch}$ and Pgk1$_{\beta11ch}$ were taken using an Airyscan microscope. Scale bar: 1 µm. (**E**) Flow cytometry measurements of total GFP fluorescence of the BiG Mito-Split-GFP strain stably expressing Pgk1$_{\beta11ch}$ or Pam16$_{\beta11ch}$ (N = 3). (**F**) The mitochondrial GatF protein was fused to the GFP$_{\beta1-10}$ fragment (mtGatF$_{\beta1-10}$), thereby targeting the ten first GFP beta-strands to mitochondria after being transcribed in the nucleus and translated in the cytoplasm. This construct was co-expressed with either cERS$_{\beta11ch}$ or Pgk1$_{\beta11ch}$. The GFP reconstitution was monitored by epifluorescence microscopy. Mitochondria were stained with MitoTracker Red CMXRos. Scale bar: 5 µm. Representative fields are shown.

The online version of this article includes the following source data and figure supplement(s) for figure 2:

**Source data 1.** Micrographs of the BiG Mito-Split-GFP expressing Pgk1$_{\beta11ch}$, cERS$_{\beta11ch}$, Pam16$_{\beta11ch}$, (related to **Figure 2B**).

**Source data 2.** Confirmation of the expression of the GFP$_{\beta1-10}$, cERS$_{\beta11ch}$ and Pgk1$_{\beta11ch}$ fusion proteins in whole cell extract from the transformed BiG Mito-Split-GFP strains (Related to **Figure 2C**).

**Source data 3.** Flow cytometry measurements of total GFP fluorescence of the three biological replicates of the BiG Mito-Split-GFP strain stably expressing Pgk1$_{\beta11ch}$ or Pam16$_{\beta11ch}$ (related to **Figure 2F**).

**Figure supplement 1.** Mitochondrial relocation of mitochondrial proteins or echoforms tagged with GFP$_{\beta11}$ (related to **Figure 2**).

or Atp4$_{\beta11ch}$ bearing the GFP$_{\beta11ch}$ tag at their C-terminus under the constitutive GPD promoter. Expression of Pam16$_{\beta11ch}$ and Atp4$_{\beta11ch}$ resulted in strong GFP signal emissions that colocalized with MitoTracker Red CMXRos-stained mitochondria, whereas no fluorescence was detected with the corresponding empty plasmid (**Figure 2B**; **Figure 2—figure supplement 1A**). These observations confirmed that the GFP$_{\beta1-10}$ polypeptide is well expressed from the mtDNA, stably and correctly folded, allowing reconstitution of an active GFP upon association with the mitochondrial GFP$_{\beta11ch}$-tagged protein. So far, the positive controls we used for the proof of concept of the BiG Mito-Split-GFP approach are proteins more or less abundant: Atp4 (30000–40000 copies/cell) and Pam16 (3000 copies/cell) (**Morgenstern et al., 2017**; **Vögtle et al., 2017**). We will report soon, in BioRxiv, tests with other proteins with a known mitochondrial location and varying abundance to better estimate the sensitivity of the BiG Mito-Split-GFP system, including the GatF subunit of the GatFAB tRNA-dependent amidotransferase chromosomally expressed from its own promoter. This is a mitochondrial protein that has been reported to be present at only 40–80 copies (**Vögtle et al., 2017**).

We next tested the BiG Mito-Split-GFP system with a GFP$_{\beta11ch}$-tagged version of Pgk1, which is commonly used as negative cytosolic marker protein to probe the purity of mitochondrial preparations. Pgk1$_{\beta11ch}$ and endogenous Pgk1 were well detected by Western blot of total protein extracts probed with anti-Pgk1 antibodies (**Figure 2C**). No GFP fluorescence was observed with Pgk1$_{\beta11ch}$ (**Figure 2B**; **Figure 2—figure supplement 1A**) despite its good expression (**Figure 2C**). This is an interesting observation considering that Pgk1 localizes at the external surface of mitochondria (**Cobine et al., 2004**; **Kritsiligkou et al., 2017**; **Levchenko et al., 2016**). This provides the proof that the BiG Mito-Split-GFP system does not yield any unspecific fluorescence with cytosolic proteins even when they are externally associated to the organelle (see also Source data 4). Another negative control (His3) that further confirms the absence of false positive signal will be provided soon in BioRxiv. In conclusion, these data show that any GFP$_{\beta11ch}$-tagged protein that localizes inside the mitochondrial matrix or at matrix side periphery of the inner membrane triggers GFP reconstitution and fluorescence emission, making this emission a robust in vivo readout for the mitochondrial importability of proteins of nuclear genetic origin.

We next tested whether the BiG Mito-Split-GFP system also allows visualization of the mitochondrial echoform of a protein located in both the cytosol and the organelle. We chose the cytosolic glutamyl-tRNA synthetase (cERS) encoded by the *GUS1* gene as a proof of concept. As we have shown, cERS is an essential and abundant protein of the cytosolic translation machinery, and a small fraction (15%) is located in mitochondria where it is required for mitochondrial protein synthesis and ATP synthase biogenesis (**Frechin et al., 2009**; **Frechin et al., 2014**). After transformation of the BiG Mito-Split-GFP strain with plasmids expressing a GFP$_{\beta11ch}$-tagged version of cERS under the control of either the GPD promoter (pGPD) or its own promoter (pGUS1), a GFP signal was observed only in

mitochondria (*Figure 2B*; *Figure 2—figure supplement 1A*). We also generated a stable BiG Mito-Split-GFP strain in which the gene encoding $cERS_{\beta11ch}$ was chromosomally expressed under the dependence of its own promoter at the *TRP1* locus (*Supplementary file 3*, *Figure 2—figure supplement 1B*). Again, GFP fluorescence was strictly confined to mitochondria (*Figure 2B*, *Figure 2—figure supplement 1A*). These observations demonstrate that the BiG Mito-Split-GFP system enables a specific detection in vivo of the mitochondrial pool of cERS ($_{mte}$cERS), without any interference by the cytosolic echoform, which is not possible when cERS is tagged with regular GFP (*Frechin et al., 2009*). We also expressed $Pam16_{\beta11ch}$ and $Pgk1_{\beta11ch}$ under the dependence of the GPD promoter at the *TRP1* locus. Again, as shown with the plasmid-borne strategy, $Pam16_{\beta11ch}$ expression resulted in a specific mitochondrial fluorescence, while $Pgk1_{\beta11ch}$ gave no fluorescence (*Figure 2—figure supplement 1B*).

Using high-resolution Airyscan confocal microscopy, a typical 3D mitochondrial network was reconstituted from the fluorescence induced by the expression of $Pam16_{\beta11ch}$, $Atp4_{\beta11ch}$ and $cERS_{\beta11ch}$ in the BiG Mito-Split-GFP strain whereas, as expected, no fluorescent at all was detected with $Pgk1_{\beta11ch}$ (*Figure 2D*), which further illustrates the mitochondrial detection specificity of this system. These data were corroborated by flow cytometry analyses of the BiG Mito-Split-GFP strain stably expressing $Pam16_{\beta11ch}$ and $Pgk1_{\beta11ch}$ (*Figure 2E*). These data will soon be completed (in BioRxiv) with flow cytometry experiments aiming to know if the BiG Mito-Split-GFP system could be used in systematic screens for proteins with a mitochondrial localization.

We next evaluated whether the BiG Mito-Split-GFP approach represents a significant technical advance compared to the existing MTS-based Split-GFP methods that are currently used. To this end, we constructed cells (with a wild type mitochondrial genome) that co-express in the cytosol the mitochondrial protein GatF (with its own MTS) fused at its C-terminus with $GFP_{\beta1-10}$ ($mtGatF_{\beta1-10}$) and either $cERS_{\beta11ch}$ (dual localized, positive control) or $Pgk1_{\beta11ch}$ (cytosolic, negative control) (*Figure 2F*, left panel). As expected, a strong and specific mitochondrial fluorescent signal was obtained with $cERS_{\beta11ch}$ (*Figure 2F*, right panel). However, $Pgk1_{\beta11ch}$ resulted in a mitochondrial signal of similar intensity. This is presumably due to the location at the external surface of mitochondria of a small fraction of the Pgk1 pool that could interact with $mtGatF_{\beta1-10}$ prior to its import into the organelle. These results show that due to the high affinity of both self-assembling Split-GFP fragments, the MTS-based strategy can generate a mitochondrial fluorescence without mitochondrial protein internalization (*Figure 2F*, right panel). These experiments suggest that compartment-restricted expression of the $GFP_{\beta1-10}$ fragment and $GFP_{\beta11ch}$-tagged proteins increases the reliability of identifying mitochondrial echoforms of dual-localized proteins.

## Screening for mitochondrial relocation of cytosolic aminoacyl-tRNA synthetases

Originally, screening cytosolic aminoacyl-tRNA synthetases (caaRSs) that can additionally relocate to mitochondria was motivated by several inconsistencies concerning this family of enzymes. The first and most documented example concerns cERS (*Frechin et al., 2009*; *Frechin et al., 2014*). We showed that the fraction of cERS which is imported ($_{mte}$cERS) into mitochondria is essential for the production of mitochondrial Gln-tRNA$^{Gln}$ by the so-called transamidation pathway (*Frechin et al., 2009*; *Frechin et al., 2014*). In the latter, $_{mte}$cERS aminoacylates the mitochondrial tRNA$^{Gln}$ with Glu thereby producing the Glu-tRNA$^{Gln}$ that is then converted into Gln-tRNA$^{Gln}$ by the GatFAB amidotransferase (AdT) (*Frechin et al., 2009*; *Frechin et al., 2014*). These results argued against the proposal that mitochondrial import of cQRS compensates for the absence of nuclear-encoded mtQRS in yeast (*Rinehart et al., 2005*). This being said, nothing excludes that cQRS can be imported into mitochondria to fulfill additional tasks beyond translation.

Another puzzling concern is the absence in *S. cerevisiae* of genes encoding six *stricto-senso* mtaaRSs: mtARS, mtCRS, mtGRS, mtHRS, mtQRS and mtVRS (*Table 2*). This suggests that the genes encoding their cytosolic equivalents ($_{cyte}$caaRS) might also encode their mitochondrial echoforms ($_{mte}$caaRSs). This has been confirmed for cARS, cGRS1, cHRS, cVRS for which alternative translation/transcription initiation allows the expression of both echoforms (*Figure 3D*; *Chang and Wang, 2004*; *Chatton et al., 1988*; *Chen et al., 2012*; *Natsoulis et al., 1986*; *Turner et al., 2000*).

We therefore applied the BiG Mito-Split-GFP strategy to the *S. cerevisiae* caaRSs (See *supplementary file 4*), aiming to discover new mitochondrial echoforms of caaRSs. We successfully expressed in the BiG Mito-Split-GFP strain the full length $GFP_{\beta11ch}$-tagged versions of 18 out of 20

Table 2. List of genes encoding *S. cerevisiae* cytosolic and mitochondrial aminoacyl-tRNA synthetases and their cytosolic or mitochondrial echoforms

| | Gene coding for | | | |
| --- | --- | --- | --- | --- |
| | aaRSs forms | | aaRS echoforms | |
| aaRS | cytosolic (c) | mitochondrial (mt) | cytosolic (cyte) | mitochondrial (mte) |
| IRS | *ILS1* | *ISM1* | - | - |
| GRS | *GRS1/GRS2* | - | *GRS1* | *GRS1 −23* |
| SRS | *SES1* | *DIA4* | - | - |
| KRS | *KRS1* | *MSK1* | - | - |
| RRS | *RRS1* | *MSR1* | - | - |
| ERS | *GUS1* | *MSE1* | *GUS1* | *GUS1* |
| VRS | *VAS1* | - | *VAS1Δ46* | *VAS1* |
| YRS | *TYS1* | *MSY1* | - | - |
| MRS | *MES1* | *MSM1* | - | - |
| NRS | *DED81* | *SLM5* | - | - |
| PRS | *YHR020W* | *AIM10* | - | - |
| TRS | *THS1* | *MST1* | - | - |
| DRS | *DPS1* | *MSD1* | - | - |
| FRS | *FRS1 (β)/FRS2 (a)* | *MSF1 (a)* | - | - |
| CRS | *CRS1* | - | - | - |
| WRS | *WRS1* | *MSW1* | - | - |
| QRS | *GLN4* | - | - | - |
| ARS | *ALA1* | - | *ALA1* | *ALA1 −25* |
| LRS | *CDC60* | *NAM2* | - | - |
| HRS | *HTS1* | - | *HTS1Δ20* | *HTS1* |

The *Saccharomyces* Genome Database standard gene names are used. The amino acid (aa) one-letter code is used for the aminoacyl-tRNA synthetase aa specificity and (-) means that the gene encoding the corresponding aaRS is missing. Two genes encode the cytosolic phenylalanyl-tRNA synthetase (cFRS) since the enzyme is an $\alpha_2\beta_2$ hetero-tetramer. For echoforms, the position of the alternative initiation start codon is indicated and corresponds to the nomenclature described in **Figure 3**; briefly, (- number) means that the start codon of the mteaaRS is located (number) aa upstream the one that starts translation of the corresponding cyteaaRS while (Δnumber) means that the start codon of the cyteaaRS is located (number) aa downstream the one that starts translation of the corresponding mteaaRS.

yeast caaRSs or cyteaaRSs (*Figure 3A–C*; *Figure 3—figure supplement 1*, *Supplementary files 3* and *4*). For unknown reasons, we failed to obtain the full-length GFP$_{\beta11ch}$-tagged versions of cCRS and cPRS despite repeated attempts, but successfully cloned the first hundred N-terminal aa residues of cCRS (N$_{100}$cCRS) (*Figure 3C*). An unambiguous mitochondrial fluorescent signal was observed with cFRS2$_{\beta11ch}$ (the $\alpha$-subunit of the $\alpha_2\beta_2$ cFRS), cytecHRS$_{\beta11ch}$ and N$_{100}$cCRS$_{\beta11ch}$ (*Figure 3A–C*; *Figure 3—figure supplement 1*). Since the existence of a fully functional mtFRS has been demonstrated (*Koerner et al., 1987*), it is possible that supernumerary mtecFRS2 we identified is not necessary for charging mitochondrial tRNA$^{Phe}$ but exerts some non-canonical functions, in addition to its role in cytosolic protein synthesis. The mitochondrial fluorescence triggered by expression of N$_{100}$cCRS$_{\beta11ch}$ suggests that this part of cCRS harbors a MTS, which has recently been proposed (*Nishimura et al., 2019*, see Discussion). The mitochondrial fluorescence triggered by cytecHRS$_{\beta11ch}$ is more intriguing. The most plausible hypothesis is that the MTS of the mtecHRS is longer than the one originally characterized. The other possibility is that there is indeed a second mitochondrial echoform of cHRS imported inside mitochondria through a cryptic MTS that has yet to be

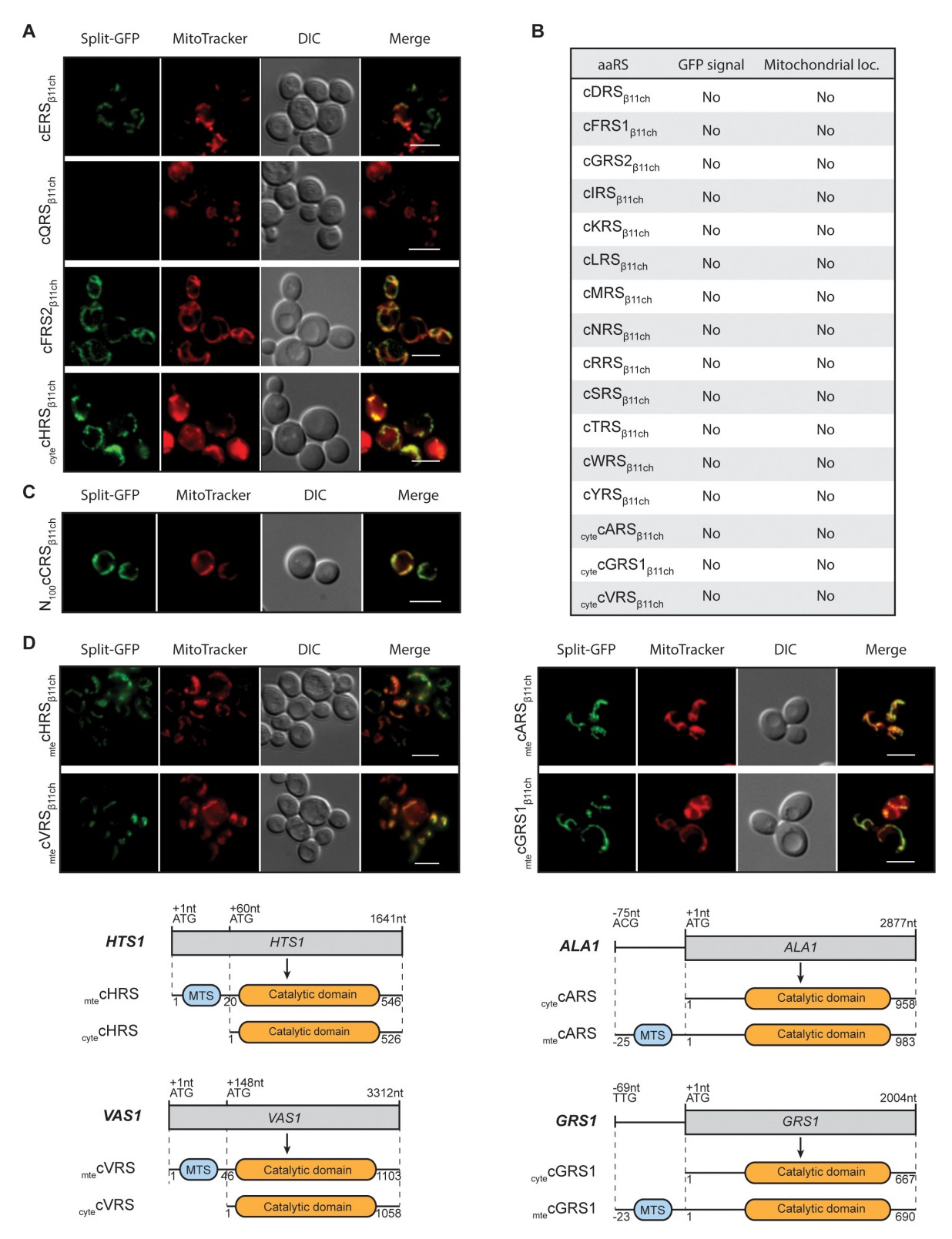

**Figure 3.** Identification and visualization of mitochondrial echoforms of yeast cytosolic aaRSs using the BiG Mito-Split-GFP strategy. Fluorescence microscopy analyses of BiG Mito-Split-GFP strain transformed with pAG414pGPD$_{\beta11ch}$ expressing yeast caaRSs (also see Table S3). Genes encoding 18 out of the 20 yeast caaRS, including those encoding the α- and β-subunits of the cytosolic α$_2$β$_2$ FRS (cFRS2), and the cGRS2 pseudogene, as well as the four encoding the cytosolic echoforms of cGRS1 ($_{cyte}$cGRS1), cARS ($_{cyte}$cARS), cHRS ($_{cyte}$cHRS) and cVRS ($_{cyte}$cVRS) were cloned in the pAG414pGPD$_{\beta11ch}$

*Figure 3 continued on next page*

*Figure 3 continued*

and expressed in the BiG Mito-Split-GFP strain (N = 2). (**A**) From the set of caaRSs tested, only cERS, cQRS, cFRS2 and $_{cyte}$cHRS micrographs are shown. (**B**) Table summarizing the GFP emission and mitochondrial localization of the caaRSs not shown in **A**. The corresponding micrographs are shown in Fig. S4A. (**C**) Fluorescence microscopy analysis of the BiG Mito-Split-GFP strain expressing the first 100 amino acids of the N-ter region of the cCRS fused to GFP$_{\beta11ch}$ (N = 2). (**D**) Fluorescence microscopy analyses of BiG Mito-Split-GFP strain transformed with pAG414pGPD$_{\beta11ch}$ expressing the mitochondrial echoforms $_{mte}$cGRS1, $_{mte}$cARS, $_{mte}$cHRS and $_{mte}$cVRS. Schematics of cARS, cGRS1, cHRS and cVRS echoforms expression in yeast. Expression can be initiated upstream of the initiator ATG$_{+1}$ ($_{mte}$cARS at ACG$_{-75}$ and $_{mte}$cGRS1 at TTG$_{-69}$) but the synthesis of this echoform can also be initiated at the ATG$_{+1}$. In this case, the expression of the cytosolic echoform is initiated downstream ($_{cyte}$cHTS at ATG$_{+60}$ and $_{cyte}$cVRS at ATG$_{+148}$). Mitochondria were stained with MitoTracker Red CMXRos. Scale bar: 5 μm. Representative fields are shown.

The online version of this article includes the following source data and figure supplement(s) for figure 3:

**Source data 1.** Confirmation, by WB, of the expression of the 18 full-length aaRS$_{\beta11ch}$ and N100cCRS$_{\beta11ch}$ in whole cell extracts from the transformed BiG Mito-Split-GFP strains (Related to *Figure 3*).

**Figure supplement 1.** Screening of caaRSs and expression level of each GFP$_{\beta11ch}$-tagged proteins (related to *Figure 3*).

identified and, like for cFRS2, this new $_{mte}$cHRS would then most probably exert a non-canonical function.

As already mentioned, cARS, cGRS1, cHRS and cVRS genes are known to produce both cytosolic and mitochondrial forms of these proteins (*Figure 3D*). When $_{mte}$cARS$_{\beta11ch}$, $_{mte}$cGRS1$_{\beta11ch}$, $_{mte}$cHRS$_{\beta11ch}$ and $_{mte}$cVRS$_{\beta11ch}$ (echoforms that start with the most upstream methionine initiator codon, *Figure 3D*) were expressed in the BiG Mito-Split-GFP strain, a mitochondrial GFP staining was, as expected, observed with these four $_{mte}$caaRSs (*Figure 3D*). Conversely, $_{cyte}$cARS$_{\beta11ch}$, $_{cyte}$cGRS1$_{\beta11ch}$ and $_{cyte}$cVRS$_{\beta11ch}$, versions without their MTS) did not produce any detectable GFP signal confirming the MTS-dependency of these cytosolic echoforms for mitochondria localization (*Figure 3D*; *Figure 3—figure supplement 1A*). The mitochondrial fluorescence produced by $_{cyte}$cHRS$_{\beta11ch}$ has already been discussed above.

## Investigating non-conventional mitochondrial targeting signals in dual localized proteins

Unlike proteins with a MTS that is cleaved upon import into mitochondria, $_{mte}$cERS does not involve any processing (*Frechin et al., 2009*). Presumably, the mitochondrial targeting residues are located in the N-terminal (N-ter) region of cERS as in precursors of mitochondrial proteins destined to the matrix. To identify them, we tagged with GFP$_{\beta11ch}$ three N-ter domains of cERS of varying length that correspond to the first 30 (cERS$_{\beta11ch}$-N1), 70 (cERS$_{\beta11ch}$-N2) and 200 (cERS$_{\beta11ch}$-N3) residues of cERS (*Supplementary files 3* and *4*; *Figure 4A*) and we tested their ability to be imported in the mitochondria of the BiG Mito-Split-GFP strain (*Figure 4B*). All three peptides produced a GFP fluorescence signal that matched the labeling of mitochondria with MitoTracker Red CMXRos (*Figure 4B*). Consistently, no GFP fluorescence was detected with cERS$_{\beta11ch}$ lacking the residues 1–30 or 1–200 (cERS$_{\beta11ch}$-ΔN1 and cERS$_{\beta11ch}$-ΔN2 respectively) (*Figure 4B*) despite detection by WB of these truncated proteins in cells (*Figure 4C*). For unknown reasons, cERS$_{\beta11ch}$-N1 and cERS$_{\beta11ch}$-N2 constructs were not detected by Western blot but gave a proper mitochondrial fluorescence staining (*Figure 4B* and **C**). These data narrow down cERS' MTS to the 30 first aa residues of its N-ter domain; this segment is made of a short β-strand and a 13 aa long α-chain (*Simader et al., 2006*) likely harboring the import signal. This further illustrates the strength of our technique towards the identification of unconventional MTSs in dual localized proteins.

## Testing mitochondrial importability of plant and mammalian proteins using the BiG Mito-Split-GFP system

The BiG Mito-Split-GFP system is based on modifications in the mitochondrial genome for expressing the GFP$_{\beta1-10}$ fragment inside the organelle. Modifying the mitochondrial genome is thus far only possible in *S. cerevisiae* and *Chlamydomonas reinhardtii* (*Remacle et al., 2006*). Owing to the high degree of conservation of mitochondrial protein import systems (*Lithgow and Schneider, 2010*), we used the yeast BiG Mito-Split-GFP strain to test the mitochondrial importability of proteins from various eukaryotic origins. We first tested two glutamyl-tRNA synthetases from *Arabidopsis thaliana*, *Ath*cERS and *Ath*mt/chlERS. According to independent MTS prediction tools, *Ath*cERS would be a cytosolic protein with a putative chloroplastic targeting signal (TargetP1.1), whereas *Ath*mt/chlERS is

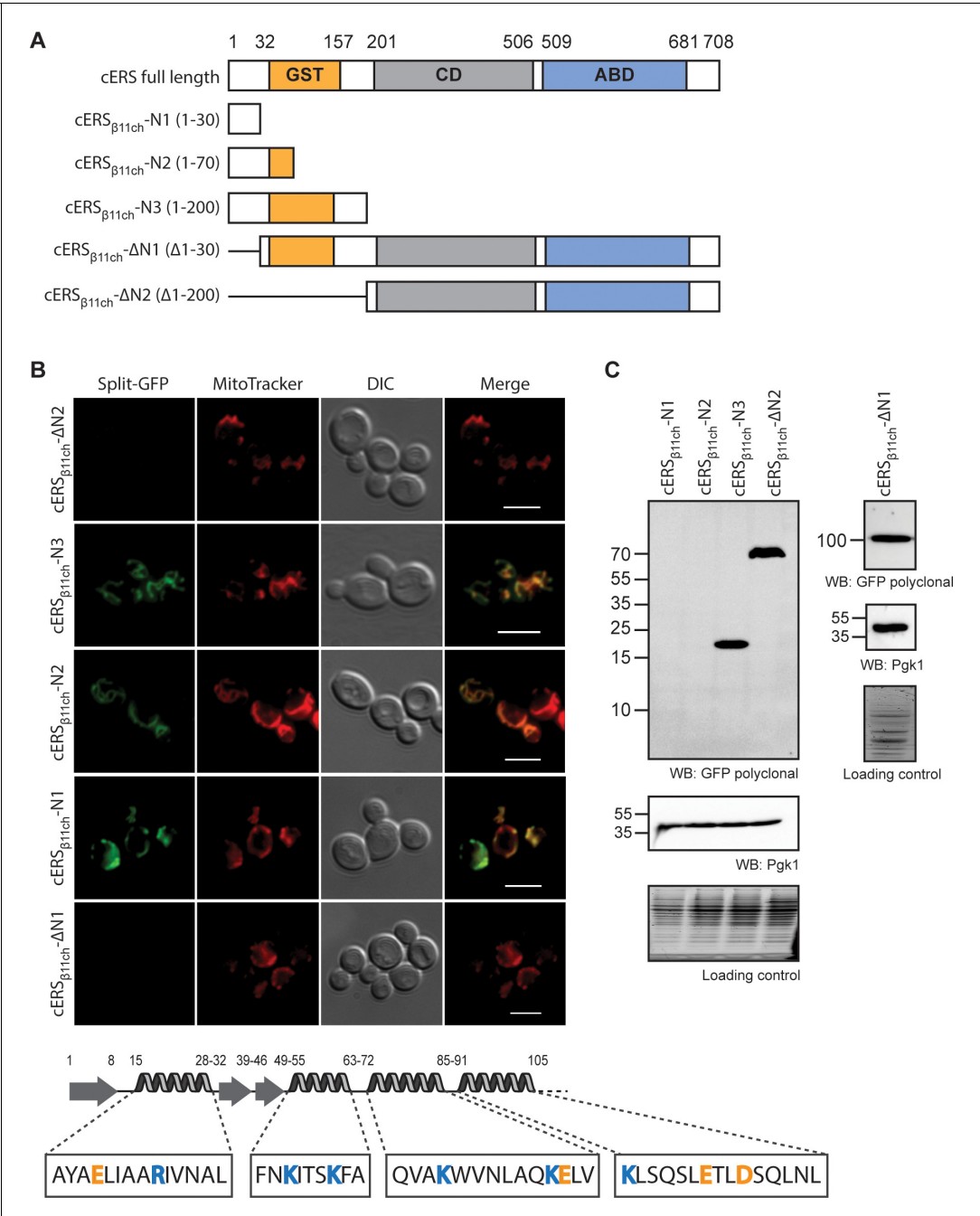

**Figure 4.** The BiG Mito-Split-GFP is a suitable tool to delimit regions containing non-canonical MTSs. (**A**) Schematic representation of the cERS fragments fused to GFP$_{\beta11ch}$. Orange boxes correspond to the GST-like domain necessary for Arc1 interaction (GST), the grey boxes represent the catalytic domain (CD), and the blue box, the tRNA-binding domain generally named anti-codon binding domain (ABD). Numbering above corresponds to cERS amino acids residues. (**B**) Fluorescence microscopy analyses of the BiG Mito-Split-GFP strain expressing the cERS variants shown on **A**. Mitochondria were stained with MitoTracker Red CMXRos; scale bar: 5 µm. The secondary structure (according to *Simader et al., 2006*) of the smallest peptide that still contains the non-conventional MTS of cERS is described together with the amino acid sequence of each helices. Positively and negatively charged amino acids are shown in orange and blue respectively. (**C**) Immunodetection of the cERS variants in BiG Mito-Split-GFP whole cell extracts using anti-GFP antibodies. Quantity of proteins loaded in each lane was estimated using anti-Pgk1 antibodies or by the stain-free procedure. The bands corresponding to the mutants N1 and N2 could not be detected. The representative fields or gel are shown.

The online version of this article includes the following source data and figure supplement(s) for figure 4:

**Source data 1.** Immunodetection of the cERS variants in BiG Mito-Split-GFP whole cell extracts using anti-GFP antibodies (related to *Figure 4C*).
**Figure supplement 1.** Analysis of N-terminal sequences of mitochondrial aaRSs and echoforms.

strongly predicted to be located in mitochondria and chloroplast (*Figure 5A*). cDNAs encoding the *Ath*cERS and *Ath*mt/chlERS proteins were fused to GFP$_{β11ch}$ (*Supplementary files 3* and *4*) and the resulting plasmids were transformed into the BiG Mito-Split-GFP strain. Expression of these proteins was confirmed by Western blot (*Figure 5C*). *Ath*cERS$_{β11ch}$ did not produce any GFP signal, whereas consistent with its predicted localization *Ath*mt/chlERS$_{β11ch}$ resulted in a specific mitochondrial fluorescence staining (*Figure 5B*). These data show that the yeast BiG Mito-Split-GFP system can be used to analyze mitochondrial localization of plant proteins.

We also used the BiG Mito-Split-GFP system to address a yet-unresolved question regarding the presence of mammalian Argonaute protein 2 (Ago2) in mitochondria. This protein mainly localizes to

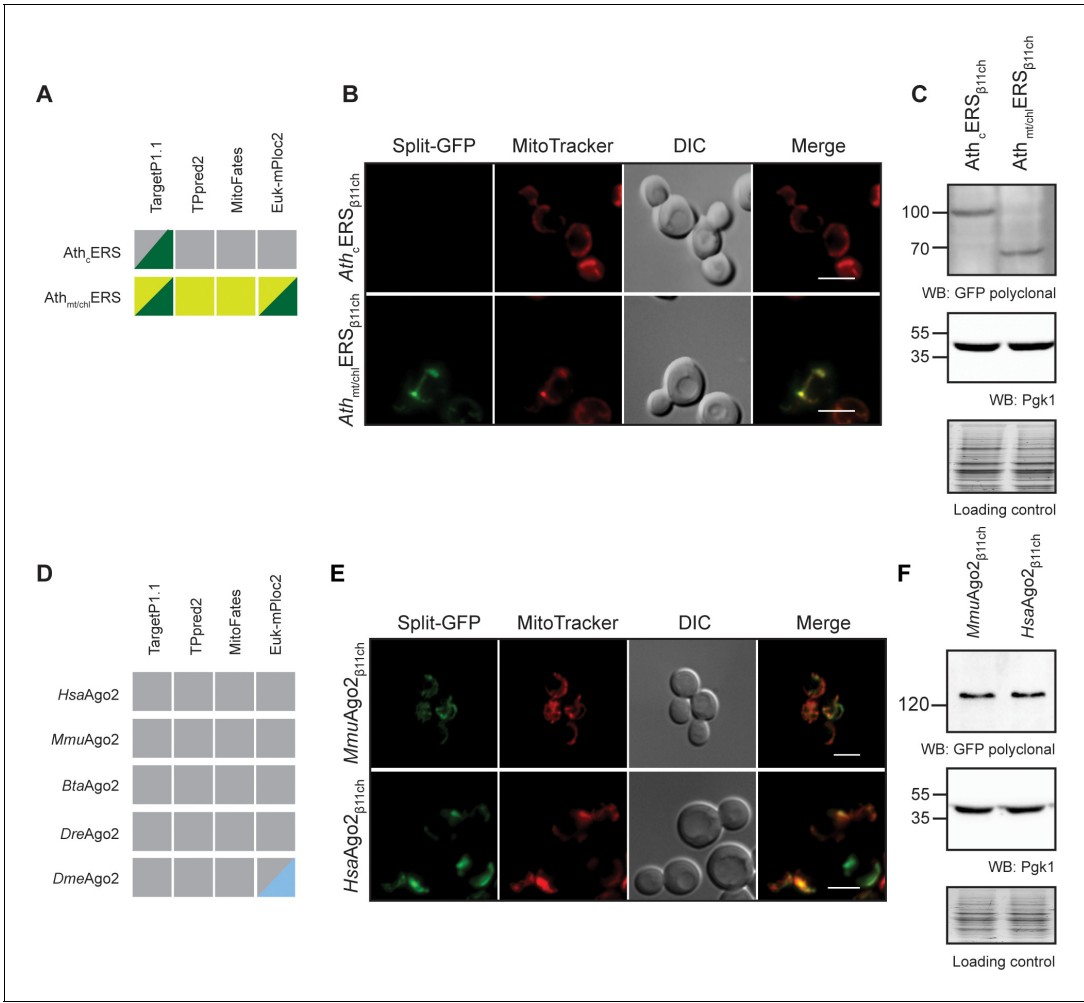

**Figure 5.** The BiG Mito-Split-GFP can be used to study mitochondrial importability of mammalian and plant proteins. (A, D) Prediction of MTS and mitochondrial localization of (A) two ERS from *Arabidopsis thaliana* (*Ath*cERS and *Ath*mt/chlERS) and (D) five eukaryotic Ago2 proteins [*Hsa*Ago2 (Protein argonaute-2 isoform X2 [Homo sapiens] NCBI sequence ID: XP_011515267.1), *Mmu*Ago2 (protein argonaute-2 *Mus musculus* NCBI sequence ID: NP_694818.3.), *Bta*Ago2 (*Bos Taurus*), *Dre*Ago2 (*Danio rerio*), *Dme*Ago2 (*Drosophila melanogaster*). MTS were predicted using TPpred2.0 (http://tppred2.biocomp.unibo.it/tppred2), TargetP1.1 (http://cbs.dtu.dk/services/TargetP/), MitoFates (http://mitf.cbrc.jp/MitoFates/cgibin/top.cgi) and the EukmPloc2 website (http://www.csbio.sjtu.edu.cn/bioinf/euk-multi-2/). Grey boxes indicate prediction of a cytosolic localization, light and dark green indicate prediction of mitochondrial or chloroplastic localization respectively. Blue boxes indicate prediction of nuclear localization. (B, E) Fluorescence microscopy analyses of the BiG Mito-Split-GFP strain expressing the GFP$_{β11ch}$-tagged $_{Ath}$cERS and $_{Ath}$mt/chlERS (N = 2) (B) and$_{Mmu}$Ago2, $_{Hsa}$Ago2 (N = 2) (E). Mitochondria were stained with MitoTracker Red CMXRos. Scale bar: 5 μm. Representative fields are shown. (C, E) Protein expression was checked by WB with anti-GFP antibodies and equal amount of loaded protein was controlled using anti-Pgk1 antibodies and by the stain-free technology (Loading control: stain-free). The representative gels are shown.

The online version of this article includes the following source data for figure 5:

**Source data 1.** Confirmation, by WB, of the expression of AthERS$_{β11ch}$ and mouse and human Ago2$_{β11ch}$ in whole cell extract from the transformed BiG Mito-Split-GFP strains (Related to *Figure 5C and F*).

the nucleoplasm and cell junctions where it is required for RNA-mediated gene silencing (RNAi) by the RNA-induced silencing complex (RISC) (*Hammond et al., 2000*). In some studies, Ago2 was suggested to be associated to mitochondria, but it remains unclear whether it localizes at the external surface or inside the organelle (*Barrey et al., 2011*; *Shepherd et al., 2017*). Using four different algorithms a potential MTS could not be predicted in Ago2 proteins from human, mouse, *Bos taurus*, *Danio rerio* and*Drosophila melanogaster*, casting doubts on the mitochondrial import of Ago2 (*Figure 5D*). To help resolve this question, the BiG Mito-Split-GFP yeast strain was transformed with plasmids expressing mouse and human Ago2$_{\beta11ch}$ proteins (*Mmu*Ago2$_{\beta11ch}$ and *Hsa*Ago2$_{\beta11ch}$, respectively, *Supplementary files 3* and *4*). Expression of each of these GFP$_{\beta11ch}$-tagged constructs was confirmed by WB, and both generated a solid and specific GFP fluorescence restricted to mitochondria (*Figure 5E and F*). These observations provide strong evidence that in addition to a cytosolic and nuclear location, Ago2 is also transported into mitochondria and is really a multi-localized protein with a mitochondrial echoform.

## Discussion

Initially designed to study protein-protein interactions and solubility, the Split-GFP technology was almost immediately hijacked to track protein localization in various cell types and compartments (*Hyun et al., 2015*; *Kaddoum et al., 2010*; *Kamiyama et al., 2016*; *Külzer et al., 2013*; *Pinaud and Dahan, 2011*; *Van Engelenburg and Palmer, 2010*). It has also been used to study the mitochondrial localization of *PARK7* upon nutrient starvation (*Calì et al., 2015*), and to detect remodeling of MERCs (mitochondria-ER contact sites) in mammalian cells (*Yang et al., 2018*). Recently, Kakimoto and coworkers developed in yeast and mammalian cells a Split-based system to analyze inter-organelles contact sites (*Kakimoto et al., 2018*). However, in these approaches both GFP$_{\beta1-10}$ and GFP$_{\beta11}$ were anchored to proteins either translated in the cytosol or following the secretory pathway. Although the latter may avoid nonspecific interaction or reconstitution of the two GFP parts, we bring herein proofs that the simultaneous synthesis of both fragments in the cytosol, coupled to their high affinity to self-assemble, may induce potential false-positive GFP emission (*Figure 2F*).

To bypass this issue, we describe herein a new and robust Split-GFP system where the first 10 segments of beta barrel GFP (GFP$_{\beta1-10}$) is expressed from the mitochondrial genome and translated inside the organelle without interfering with mitochondrial function (*Figure 1C and D*). The remaining beta barrel is concatenated (GFP$_{\beta11ch}$), tagged to the protein of interest and expressed from cytosolic ribosomes. As a result, any detected GFP fluorescence obligatory originates from the organelle thereby demonstrating a mitochondrial localization for the tested proteins (*Figure 6A–B*).

This system was first successfully tested with two mitochondrial proteins (Atp4 and Pam16), and a cytosolic one (Pgk1) as a negative control. Moreover, the mitochondrial echoform of the cytosolic glutamyl-tRNA synthetase ($_{mte}$cERS) encoded by the *GUS1* nuclear gene was also detected with the BiG Mito-Split-GFP system (*Figures 2*, *3*, *4* and *6A*). As we already showed, synchronous release of cERS and cMRS from the cytosolic anchor Arc1 protein is required for a coordinated expression of mitochondrial and nuclear ATP synthase genes (*Frechin et al., 2009*; *Frechin et al., 2014*). Mitochondrial relocation of cERS is consistent with the functional plasticity of caaRSs with multiple locations in cells. Using GFP$_{\beta11ch}$-tagged N-ter segments of cERS, we localized its cryptic MTS within the first 30 aa residues. This region lacks amphiphilic residues (residues 15–28) and folds into a β-strand-loop-α−helix motif different than regular MTSs (*Roise et al., 1988*; *Simader et al., 2006*; *Figure 4*). These findings demonstrate that the BiG Mito-Split-GFP system allows not only to visualize in living cells the mitochondrial pool of proteins with multiple cellular locations, but also to decipher their non-conventional MTSs.

Recent efforts made to identify mitochondrial proteins and assign their submitochondrial localization revealed an exquisite precision (*Morgenstern et al., 2017*). However, resolving mitochondrial proteomes is challenging due to the difficulty of obtaining pure mitochondria and because many proteins transiently localize in mitochondria and are found elsewhere in cells. Up to 10–20% of the yeast mitoproteome was suggested to be composed of proteins with another location in cells (*i.e* the cytosol, the nucleus, ER…) (*Ben-Menachem and Pines, 2017*; *Morgenstern et al., 2017*). Our BiG Mito-Split-GFP system will be especially helpful to resolve these proteome complexities. This system was here applied to proteins involved in tRNA aminoacylation, some of which are well-known to relocate in different compartment to fulfill a wide range of cellular activities (*Han et al., 2012*;

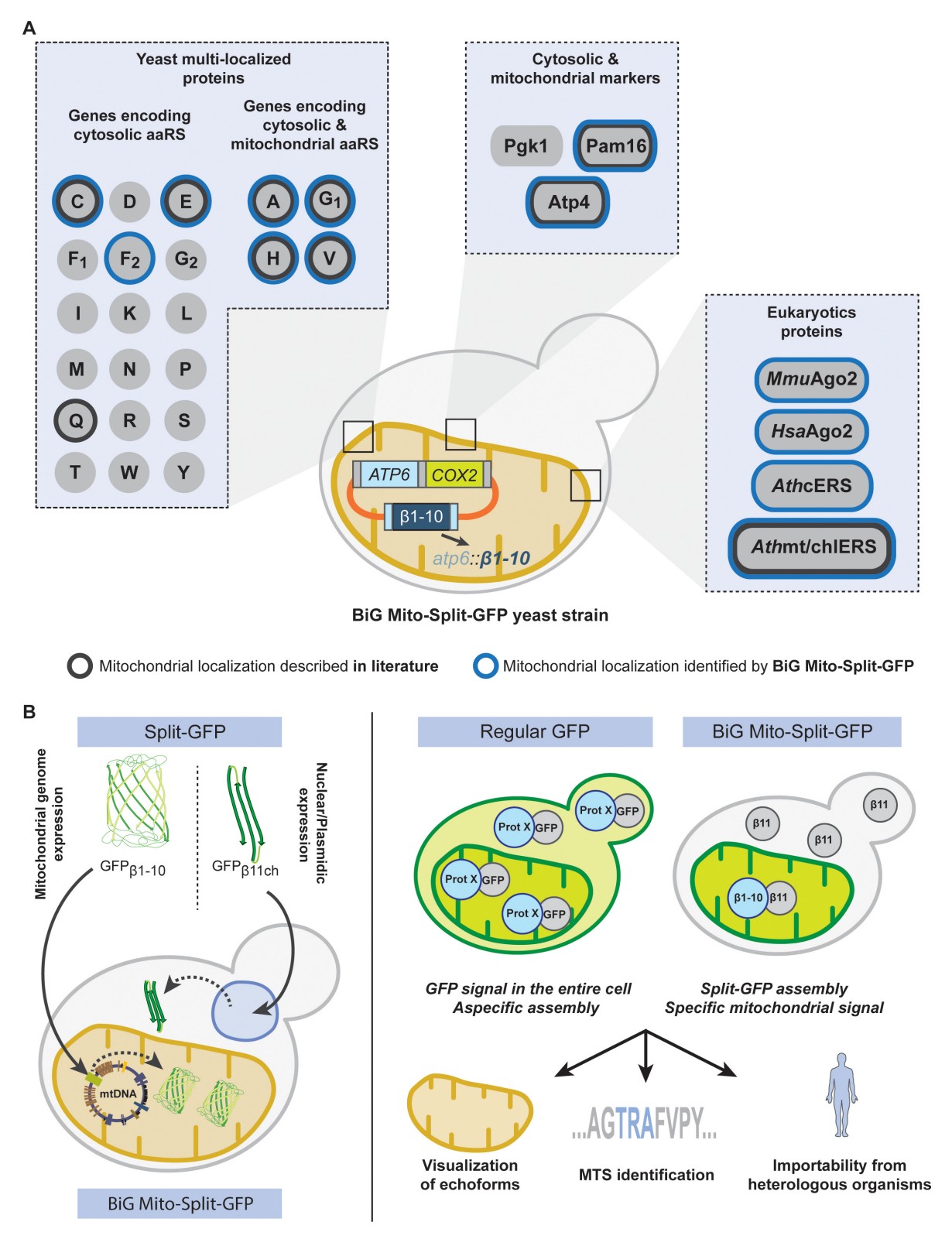

**Figure 6.** Schematic of the BiG Mito-Split-GFP system and its applications. (**A**) Using our engineered strain, we could show the dual localization of echoforms in the aaRS family of proteins and foster its power by studying localization of heterologous proteins originating from plants, mice and human. (**B**) The BiG Mito-Split-GFP strain was generated by integrating the sequence encoding the first 10 beta barrel segments into yeast mitochondrial DNA, and by either expressing any protein of interest fused to the 11th GFP segment from a plasmid or by integration in yeast nuclear

*Figure 6 continued on next page*

Figure 6 continued

DNA. As opposed to regular GFP-tagging where visualizing an echoform ultimately results in a GFP signal diffusing in the entire cell, our BiG Mito-Split-GFP system abolishes the fluorescence originating from cytosolic echoform to only display a specific mitochondrial signal. Further applications range from high-throughput experiments to identify relocating proteins involved in mitochondria homeostasis or metabolism, to identify non-conventional MTSs or seek for mitochondrial localization of heterologous proteins.

*Ko et al., 2000*; *Yakobov et al., 2018*). In this way, we provide strong evidence that cFRS2 and $_{cyte}$cHRS are dual localized as was observed for cERS, which suggests that these proteins may have additional roles beyond translation (*Figure 6A*). Being dually localized in the cytosol and mitochondria, and since there is no $_{mte}$cFRS1, it can be inferred that the catalytic α-subunit (cFRS2) is not inevitably in complex with the β-subunit within the $α_2β_2$ heterotetrameric form of cFRS. It will be interesting to test whether these findings in yeast extend to heterotetrameric cFRS from other eukaryotes, including humans. A *bona fide* mtFRS (encoded by the *MSF1* gene) that was shown to function as a monomer is essential to generate mitochondrial Phe-tRNA$^{Phe}$ (F-$_{mt}$tRNA$^F$) in mitochondria (*Sanni et al., 1991*). This further supports the hypothesis that $_{mte}$cFRS2 is not required to produce F-$_{mt}$tRNA$^F$ but more likely has a non-canonical yet-to-be-discovered function. Our failure to detect a mitochondrial echoform for cQRS is consistent with our previous findings (*Frechin et al., 2009*) that the only source of Q-$_{mt}$tRNA$^Q$ in mitochondria is provided by the relocation of $_{mte}$cERS into the organelle (*Figure 3* and *Figure 3—figure supplement 1*) *de concert* with the tRNA-dependent Gat-FAB Adt (*Frechin et al., 2014*). This definitely casts in doubt the previous proposal of the existence of a cQRS mitochondrial echoform (*Rinehart et al., 2005*). In agreement with our results (*Figure 3C*), mitochondrial echoforms of cCRS were also detected in a recent study and shown to result from alternative transcription and translation starts (*Nishimura et al., 2019*), thereby unraveling how mtCRS is expressed from the *CRS1* gene and rationalizing how mitochondrial Cys-tRNA$^{Cys}$ is produced.

Having identified new mitochondrial echoforms of caaRSs, we wondered if they carry in their N-terminal regions some common specific sequence or structural features possibly driving mitochondrial import. No specific motif was found using MAST/MEME analysis (*Bailey et al., 2009*), and there was no significant sequence similarity (as tested with Blast) (*Figure 4—figure supplement 1*). All but $_{mte}$cARS show at least one α−helix within their 50 first aa residues, and most (except cERS) are enriched in positively- vs negatively-charged aa residues, as in classical mitochondrial targeting sequences. Due to the lack of 3D structures, we cannot rule out that these N-termini adopt some specific ternary structure that are important for mitochondrial localization. As we have shown, most of the cytosolic form of cERS interacts with Arc1 in fermenting yeast, but during the diauxic shift, Arc1 expression is repressed, allowing the generation of a free pool of cERS able to relocate into mitochondria. Thus, in the case of this caaRS, interactions of its N-terminal domain seem to be important to distribute it between the cytosol and mitochondria. Future work is required to know whether such a mechanism operates also for the other dually localized caaRSs.

Our BiG Mito-Split-GFP system requires modifications of the mitochondrial genome, which can be achieved in only a limited number of organisms (*S. cerevisiae* **Bonnefoy and Fox, 2001** and *C. Reinhardtii* **Remacle et al., 2006**). However, due to the good evolutionary conservation of mitochondrial protein import, we reasoned that the system we developed in yeast could be used to test proteins of various eukaryotic origins, and we present evidence that this is indeed the case (*Figure 5*; *Figure 6C*). For instance, we showed that the mammalian Ago2 protein (*Hsa-* and *Mmu*Ago2, *Figure 5*) heterologously-expressed in yeast localize inside mitochondria. This protein was suggested to be exclusively located at the external surface of mitochondria in human cells where it would help the transport of pre- and miRNAs into the organelle, as do numerous nuclear-encoded pre- and miRNAs (*Bandiera et al., 2011*; *Barrey et al., 2011*; *Kren et al., 2009*). Several studies have suggested that mitochondrial miRNAs, also termed mitomiRs, play a role in apoptosis (*Kren et al., 2009*), mitochondrial functions (*Das et al., 2012*), and translation (*Bandiera et al., 2011*; *Jagannathan et al., 2015*; *Li et al., 2016*; *Zhang et al., 2014*), and this would require the mitochondrial import of Ago2 (*Bandiera et al., 2011*; *Das et al., 2012*; *Jagannathan et al., 2015*; *Li et al., 2016*; *Zhang et al., 2014*). However, the import of mitomiRs is still poorly understood and several possible import mechanisms have been evoked (*Barrey et al., 2011*; *Shepherd et al., 2017*). Our unambiguous detection

of Ago2 inside mitochondria of yeast cells expressing this protein sheds new light on its potential role in miRNAs import.

The yeast BiG Mito-Split-GFP system we describe here is designed to point out mitochondrial echoforms. It is robust, not expensive and can be used to test proteins from various organisms. This new approach has certainly many potential applications and opens new avenues in the study of mitochondria and their communications with other compartments of the cell.

# Materials and methods

## Key resources table

| Reagent type (species) or resource | Designation | Source or reference | Identifiers | Additional information |
|---|---|---|---|---|
| Genetic reagent (*S. cerevisiae*) | BiG Mito- Split-GFP | This study | | RKY176 strain with *ADE2* gene ($\rho^+$ atp6::GFP$_{\beta1-10}$ 5'UTR$_{COX2}$ ATP6 3'UTR$_{COX2}$) |
| Genetic reagent (*S. cerevisiae*) | BiG Mito- Split-GFP+Pgk1$_{\beta11ch}$ | This study | | RKY176 strain (*PGK1:: β11ch::TRP1*) |
| Genetic reagent (*S. cerevisiae*) | BiG Mito- Split-GFP+PAM16$_{\beta11ch}$ | This study | | RKY176 strain (*PAM16:: β11ch::TRP1*) |
| Genetic reagent (*S. cerevisiae*) | BiG Mito- Split-GFP+cERS$_{\beta11ch}$ | This study | | RKY176 strain (*GUS1:: β11ch::TRP1*) |
| Antibody | Anti-GFP (Mouse polyclonal) | Sigma | Cat# G1544 | WB (1:5000) Called GFP N-ter in *Figure 2C* recognizes GFPβ1–10 |
| Antibody | Anti-GFP (Mouse monoclonal IgG$_1\kappa$ clones 7.1 and 13.1) | Roche | Cat# 11814460001 | WB (1:5000) Called GFP polyclonal in *Figure 2C* recognizes GFPβ11 |
| Antibody | Anti-Pgk1 (Mouse monoclonal IgG1, clone 22C5D8) | Molecular Probes | Cat# 459250 | WB (1:5000) |
| Recombinant DNA reagent | pAG414-p*GPD*-β11ch (plasmid) | This study | | Template vector used for all constructs. Cloning done by Gibson assembly |
| Chemical compound, drug | MitoTracker Red CMXRos | ThermoFisher | Cat# M7512 | Mitochondria staining |
| Chemical compound, drug | 0.5% (v/v) 2,2,2-Trichloroethanol | Sigma | Cat# T54801 | Used to detect total protein loading in SDS-PAGE, referred to Loading control |

## Construction of plasmids

*ATP6* gene flanked by 75 bp of 5'UTR and 118 bp of 3'UTR of *COX2* was synthesized by Genescript and cloned at the EcoRI site of pPT24 plasmid bearing the sequence of *COX2* gene along with its UTRs (*Thorsness and Fox, 1993*), giving pRK49-2. The *GFP$_{\beta1-10}$* sequence (optimized for mitochondrial codon usage) encoding the first ten β-strands of GFP flanked by the regulatory sequences of *ATP6* gene and BamHI/EcoRI sites was synthesized by Genescript. The BamHI-EcoRI DNA fragment was cloned into pPT24 plasmid, giving the pRK67-2. The sequences of inserts were verified by sequencing.

The *GFP$_{\beta11ch}$* coding sequence, synthesized by Genescript, was subcloned into the pAG414 pGPD-ccdB vector to generate the pAG414pGPD-ccdB$_{\beta11ch}$. All genes encoding cytosolic or mitochondrial proteins were amplified from genomic DNA using the PrimeSTAR Max polymerase according to the manufacturer instructions (Takara), purified by PCR clean up (Macherey-Nagel) and subcloned either by Gateway (Thermofisher) (*Katzen, 2007*) or Gibson assembly (NEB)

(*Gibson et al., 2010*; *Gibson et al., 2009*) according to the manufacturer's instructions (see Table S2).

## Construction of the BiG Mito-Split-GFP strain

The genotypes of strains used in this study are listed in *Table 1*. The $\rho^+$ indicates the wild-type complete mtDNA (when followed by deletion/insertion mutation it means the complete mtDNA with a mutation). The $\rho^-$ synthetic genome ($\rho^{-S}$) was obtained by biolistic introduction into mitochondria of $\rho^0$ DFS160 strain (devoid of mitochondrial DNA) of the plasmids (pRK49-2 or pRK67-2) bearing indicated genes. The integration of *ATP6* gene into the mtDNA under the control of regulatory sequences of *COX2* was done using a previously described procedure (*Steele et al., 1996*). The pRK49-2 plasmid was introduced into mitochondria of DFS160 $\rho^0$ strain by ballistic transformation using the Particle Delivery Systems PDS-1000/He (*BIO-RAD*) as described (*Bonnefoy and Fox, 2001*), giving the $\rho^{-S}$ strain RKY89. For the integration of the *ATP6* gene at the *COX2* locus, we first constructed a $\rho^+$ strain (RKY83, Fig. S2A) with a complete deletion of the coding sequence of *ATP6* (atp6::ARG8m) and a partial deletion in *COX2*, cox2-62 (*Table 1*), by crossing YTMT2 (Matα derivative of strain NB40-3C carrying the cox2-62 mutation (*Steele et al., 1996*) and MR10 (atp6::ARG8m) (*Rak et al., 2007*). After crossing, cells were allowed to divide during 20–40 generations to allow mtDNA recombination and mitotic segregation of the double mutation. The double atp6::ARG8m cox2-62 mutant colonies were identified by crossing with the $\rho^{-S}$ strain SDC30 (*Duvezin-Caubet et al., 2003*) that carries *ATP6* and *COX2* which restored the respiratory competence and by crossing with the YTMT2 strain, $\rho^+$cox2-62, which did not restored the respiratory competence of the double mutant. Next, the $\rho^{-S}$ strain RKY89 was crossed with strain RKY83. This cross resulted in the respiratory competent progenies, named RKY112, which were growing on minimal medium without arginine (*Table 1*, *Figure 1B* and S2B). The ectopic integration of the *ATP6* gene into *COX2* locus was verified by PCR using oligonucleotides oAtp6-2, oAtp6-4, o5'UTR2 and o5'UTR1 (Table S1, Fig. S2D). To integrate $GFP_{\beta1-10}$ into *ATP6* locus the $\rho^{-S}$ strain RKY172 was obtained by biolistic transformation of DFS160$\rho^0$ with pRK67-2, as described above. RKY172 was crossed with RKY112, heterokaryons were allowed to divide during 20–40 generations to allow mtDNA recombination and mitotic segregation (Fig. S2C). The RKY176 progenies were selected by their respiratory competence and inability to grow on arginine depleted plates. The correct integration of the $GFP_{\beta1-10}$ gene into *ATP6* locus was verified by PCR using oligonucleotides oAtp6-1, oAtp6-10, oXFP-pr and oXFP-lw (Table S1, Fig. S2E). Finally, *ADE2* WT sequence was amplified from the genomic DNA of a BY strain using oligonucleotides *ADE2* Fw and *ADE2* Rv (Table S2) and transformed into the RKY176 strain. Red/white colonies were then screened on adenine depleted plates to select *ADE2*-bearing RK176 strain.

## Media and growth conditions

Yeast cell culture media and their composition: complete glucose YP medium (1% Bacto yeast extract, 1% Bacto peptone, 2% glucose, 40 mg/l adenine), complete YP Gal (1% Bacto yeast extract, 1% Bacto peptone, 2% galactose, 40 mg/l adenine), synthetic media composed of 0.67% (w/v) yeast nitrogen base without amino acids (aa), 0.5% (w/v) ammonium sulfate, either 2% (w/v) glucose (SC), galactose (SC Gal) or glycerol (SC Gly) and a mixture of aa and bases from Formedium (Norfolk, UK). Low sulfate medium LSM contained 0.67% (w/v) yeast nitrogen base without aa and ammonium sulphate, 2% galactose and 50 mg/L histidine, tryptophan, leucine, uracil, adenine, and arginine. The solid media contained 2% (w/v) of agar. Every strain was grown at 30°C with rotational shaking to mid-log ($OD_{600\ nm}$ = 0.7). SC Gal was filtered on 25 µm filters and not autoclaved before use.

## Pulse-labelling of mitochondrially-synthesized proteins and ATP synthesis

Labeling of mitochondrial translation products was performed using the protocol described by *Barrientos et al., 2002*. Yeast cells were grown to early exponential phase ($10^7$ cells/mL) in 10 mL of liquid YP Gal medium. Cells were harvested by centrifugation and washed twice with LSM medium then suspended in the same medium and incubated for cysteine and methionine starvation for 2 hr at 28°C with shaking. Cells were suspended in 500 µL of LSM medium, and 1 mM cycloheximide was added. After a 5 min incubation at 28°C, 0.5 mCi of [$^{35}$S]methionine and [$^{35}$S]cysteine (Amersham

Biosciences) was added and cell suspension was further incubated for 20 min at 28°C. Total proteins were isolated by alkaline lysis and suspended in 50 µL of Laemmli buffer. Samples with the same level of incorporated radioactivity were separated by SDS-PAGE in 17.5% (w/v) acrylamide gels (to separate Atp8 and Atp9) or 12% (w/v) acrylamide containing 4 M urea and 25% (v/v) glycerol (to separate Atp6, Cox3, Cox2 and cytochrome b). After migration, the gels were dried and [$^{35}$S]-radio-labeled proteins were visualized by autoradiography with a PhosphorImager after a one-week expo-sure. To measure ATP synthase activities in the RKY112 strain, mitochondria were prepared by the enzymatic method as described in *Guérin et al., 1979*. For the rate of ATP synthesis, the mitochon-dria (0.15 mg/mL) were placed in a 1 mL thermostatically controlled chamber at 28°C in respiration buffer (0.65 M mannitol, 0.36 mM EGTA, 5 mM Tris-phosphate, 10 mM Tris-maleate pH 6.8) (*Rigoulet and Guerin, 1979*). The reaction was started by adding 4 mM NADH and 750 µM ADP; 100 µL aliquots were taken every 15 s and the reaction was stopped by adding 3.5% (v/v) perchloric acid and 12.5 mM EDTA. Samples were neutralized to pH 6.5 by KOH and 0.3 M MOPS. ATP was quantified using the Kinase-Glo Max Luminescence Kinase Assay (Promega) and a Beckman Coulter's Paradigm Plate Reader.

## Flow cytometry analysis

5 mL of cells stably expressing Pam16$_{\beta11ch}$ and Pgk1$_{\beta11ch}$ strains (see *Table 1*) grown in YPD to con-fluence were diluted in 4 mL of SC Gal and grown overnight to reach mid-log phase. They were then diluted again in SC Gal and grown for 6 hr. Cells were then centrifuged and resuspended in water, passed for GFP detection on a BD FACS Canto II cytometer and Data analysis was performed using FlowJo.

## Proteins extraction and western blots

10 mL of cells grown to mid-log phase were harvested and spin down 5 min at 2000 × *g* at room temperature (RT). Cells were suspended in 500 µL of deionized water, 50 µL of 1.85 M NaOH was added and the mixture was incubated 10 min on ice. After addition of 50 µL of TCA 100% and 10 min of incubation on ice, the total precipitate was pelleted by centrifugation 15 min at 13000 × *g* at 4°C. After removing the supernatant, pellets were suspended in 200 µL of Laemmli buffer (1×) sup-plemented with 20 µL of 1M Tris Base pH 8.

For each strain, 10 µL of total proteins were separated by SDS-PAGE on 8-, 10- or 12% (w/v) poly-acrylamide gels prior to electroblotting with a Trans-Blot Turbo system (*BIO-RAD*) onto PVDF mem-branes (*BIO-RAD*, #1704156). Detection was carried out using mouse monoclonal IgG anti-GFP primary antibodies (1:5000; Roche Clone 7.1 and 13.1) + mouse polyclonal for the recognition of GFP$_{\beta1-10}$ (1:5000, Sigma #G1544), and mouse monoclonal IgG1 anti-Pgk1 primary antibodies (1:5000; Molecular Probes Clone 22C5D8). Secondary antibodies were Goat anti-mouse IgG (H+L) HRP-conjugated antibodies (*BIO-RAD*; #1706516), at a concentration of 1:10000. ECL-plus reagents (*BIO-RAD*) was used according to the manufacturer's instructions and immuno-labeled proteins were revealed using a ChemiDoc Touch Imaging System (*BIO-RAD*). Total load of protein (Loading con-trol) was assessed by UV detection using a ChemiDoc Touch Imaging System (*BIO-RAD*; Stain-free procedure) and detected by addition of 0.5% (v/v) 2,2,2-Trichloroethanol (Sigma #T54801) to the 30% acrylamide/bis-acrylamide solution.

## Image acquisition and staining

Cells were incubated overnight in the appropriate media, diluted to an OD$_{600\ nm}$ of 0.3 prior to microscopy studies and stained after 6 hr of growth at 30°C. For mitochondria staining, cells were centrifuged 1 min at 1500 × *g* at room temperature, suspended in 1 mL of SC Gal supplemented with Red-Mitotracker CMXRos at a final concentration of 100 nM (Molecular Probes), and incubated 15 min at rotational shaking at 30°C. Cells were washed three times in one volume of deionized water, and suspended in 100 µL of deionized water for microscopic studies. Epifluorescence images were taken with an AXIO Observer d1 (Carl Zeiss) epifluorescence microscope using a 100 × plan apochromatic objective (Carl Zeiss) and processed with the Image J software. Images for 3D recon-struction were taken using a confocal LSM 780 high resolution module Airyscan with a 63 × 1.4 NA plan apochromatic objective (Carl Zeiss) controlled by the Zen Black 2.3 software (Carl Zeiss). Z-stack reconstruction was performed on the IMARIS 9.1.2 (Bitplane AG) software.

## Acknowledgements

We first thank Elodie Vega (Plateau d'imagerie cellulaire I2MC Toulouse INSERM UMR1048 – TRI Génotoul) for technical help on Airyscan images acquisition and 3D reconstruction. We also thank Laurence Huck and Maximilien Geiger for their technical assistance. The work was supported by the French National Program Investissement d'Avenir administered by the ''Agence National de la Recherche'' (ANR), ''MitoCross'' Laboratory of Excellence (Labex), funded as ANR-10-IDEX-0002–02 (to HDB, GB, LE, MH, YA, YOC, BS), the University of Strasbourg (HDB, GB, LE, MH, YA, YOC, BS, SP, SF), the CNRS (HDB, GB, LE, MH, YA, YOC, BS, SP, SF); the National Science Center of Poland grant nr UMO-2018–31-B-NZ3-01117 and UMO-2011-01-B-NZ1-03492 (to RK); the Japanese Society for Promotion of Science (JSPS) Postdoctoral Fellowship for Research Abroad and Naito Foundation (to YA); the Ministère de l'Education Nationale, de la Recherche et de l'Enseignement Supérieur (GB, LE, MH), NIH R01 5R01GM111873-02 (to J-P di R) and from the Swiss National Science Foundation and the Canton of Basel (to JP).

## Additional information

### Funding

| Funder | Grant reference number | Author |
|---|---|---|
| Agence Nationale de la Recherche | ANR-10-IDEX-0002-02 | Gaétan Bader<br>Ludovic Enkler<br>Yuhei Araiso<br>Marine Hemmerle<br>Bruno Senger<br>Hubert Dominique Becker |
| National Science Centre of Poland | UMO-2018-31-B-NZ3-01117 | Roza Kucharczyk |
| National Science Centre of Poland | UMO-2011-01-B-NZ1-03492 | Roza Kucharczyk |
| NIH R01 | 5R01GM111873-02 | Jean-Paul di Rago |
| AFM-Téléthon | N°21809 | Sylvie Friant |
| Swiss National Science Foundation | | Jean Pieters |
| University of Strasbourg | | Gaétan Bader<br>Ludovic Enkler<br>Yuhei Araiso<br>Marine Hemmerle<br>Bruno Senger<br>Sylvie Friant<br>Hubert Dominique Becker |
| Centre National de la Recherche Scientifique | | Gaétan Bader<br>Ludovic Enkler<br>Yuhei Araiso<br>Marine Hemmerle<br>Bruno Senger<br>Sylvie Friant<br>Hubert Dominique Becker |
| Ministry of Higher Education, Research and Innovation | | Gaétan Bader<br>Ludovic Enkler<br>Marine Hemmerle |
| Japan Society for the Promotion of Science | Postdoctoral Fellowship for Research Abroad | Yuhei Araiso |

The funders had no role in study design, data collection and interpretation, or the decision to submit the work for publication.

## Author contributions
Gaétan Bader, Conceptualization, Resources, Data curation, Formal analysis, Validation, Investigation, Methodology, Writing - original draft; Ludovic Enkler, Conceptualization, Resources, Data curation, Formal analysis, Investigation, Writing - original draft, Writing - review and editing; Yuhei Araiso, Conceptualization, Data curation, Formal analysis, Supervision, Validation, Investigation, Visualization, Methodology, Writing - original draft; Marine Hemmerle, Resources, Data curation, Formal analysis, Validation, Investigation, Visualization, Methodology, Writing - original draft; Krystyna Binko, Emilia Baranowska, Data curation, Formal analysis, Validation, Investigation, Visualization, Methodology; Johan-Owen De Craene, Conceptualization, Formal analysis, Supervision, Validation, Visualization, Writing - review and editing; Julie Ruer-Laventie, Data curation, Formal analysis, Validation, Investigation, Visualization, Methodology, Writing - original draft; Jean Pieters, Resources, Data curation, Supervision; Déborah Tribouillard-Tanvier, Formal analysis, Writing - review and editing; Bruno Senger, Conceptualization, Data curation, Formal analysis, Supervision, Validation, Investigation, Writing - review and editing; Jean-Paul di Rago, Conceptualization, Data curation, Formal analysis, Supervision, Validation, Writing - original draft, Writing - review and editing; Sylvie Friant, Data curation, Formal analysis, Validation, Investigation, Visualization, Writing - original draft, Writing - review and editing; Roza Kucharczyk, Conceptualization, Data curation, Formal analysis, Supervision, Funding acquisition, Validation, Investigation, Visualization, Writing - original draft, Project administration, Writing - review and editing; Hubert Dominique Becker, Conceptualization, Data curation, Formal analysis, Supervision, Funding acquisition, Validation, Writing - original draft, Project administration, Writing - review and editing

## Author ORCIDs
Roza Kucharczyk (ID) https://orcid.org/0000-0002-8712-7535
Hubert Dominique Becker (ID) https://orcid.org/0000-0002-4102-7520

## Decision letter and Author response
Decision letter https://doi.org/10.7554/eLife.56649.sa1
Author response https://doi.org/10.7554/eLife.56649.sa2

# Additional files

### Supplementary files
• Supplementary file 1. Sequence of the BamHI-EcoRI DNA fragment of GFP$_{\beta1-10}$ flanked by the regulatory sequences of *ATP6* gene Regulatory sequences of ATP6 are underlined, 5'-BamHI and 3'-EcoRI sites are in italicized bold characters. The GFP$_{\beta1-10}$ sequence is in gray background and has been codon-optimized to be expressed by *S. cerevisiae* mitochondrial translation machinery.

• Supplementary file 2. Primers used in the study to verify integration of ectopic *ATP6* or *GFP$_{\beta1-10}$* in mtDNA. The use of each oligo is described in the Materials and methods section.

• Supplementary file 3. Primers used for PCR amplifications of genes fused to GFP$_{\beta11ch}$ sequence. The primers in black and blue were used for Gateway and Gibson cloning methods respectively (see Material and methods section).

• Supplementary file 4. List of expression plasmids generated for this study.

• Transparent reporting form

### Data availability
Source data for all figures showing blots and microscopy images have been provided.

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
