## [Decision Letter]

**Acceptance summary:**

We are excited to publish this paper as we feel that this work describes a long awaited, "ultimate" version of the split-GFP technique for the study of mitochondrial import. The presented data clearly shows that the method works and is widely applicable in the field of mitochondrial biology. The work presents a masterful use of yeast genetics and makes a very significant contribution to the field.

**Decision letter after peer review:**

Thank you for submitting your article "Assigning mitochondrial localization of dual localized proteins using a yeast Bi-Genomic Mitochondrial-Split-GFP" for consideration by *eLife*. We are happy to say that we find your article suitable for publication following some required revisions.

Your article has been reviewed by three peer reviewers, one of whom is a member of our Board of Reviewing Editors, and the evaluation has been overseen by Dominique Soldati-Favre as the Senior Editor. The reviewers have opted to remain anonymous.

The reviewers have discussed the reviews with one another and the Reviewing Editor has drafted this decision to help you prepare a revised submission.

Please find below a summary of the points agreed upon by the reviewers and the reviewing editor:

Summary

The work by Bader et. al presents a new bigenomic fluorescent complementation reporter (BiG Mito-Split-GFP) for assessing the mitochondrial localization of dually localized proteins. First, the authors describe the creation of the reporter. Using an intricate yeast genetics approach the authors integrated a larger part (β-sheets 1-10) of the split-GFP coding sequence into the yeast mitochondrial genome. The smaller part of the split-GFP (β-sheet 11) was fused to a number of studied proteins on plasmids. Second, the authors demonstrate that the reporter correctly shows the localization of known mitochondrial proteins and gives no/little signal for a protein which is only cytosolic. The described BiG Mito-Split-GFP reporter is compared with an available method in which both split-GFP components are encoded in the nuclear genome and the BiG Mito-Split-GFP is shown to be superior. Then the authors demonstrate the application of their technique to the study of mitochondrial echoforms of amino acid tRNA synthetases (aaRSs). They discover a new echoform for phenylalanine aaRS and look into the targeting signal for glutamate aaRS mitochondrial echoform. Finally, the application for the study of mitochondrial import of heterologous proteins from animals, plants and algae is shown. For instance, the authors demonstrate that Ago2 protein from mammals has a capacity to be imported into yeast mitochondria. This work describes a long awaited, "ultimate" version of the split-GFP technique for the study of mitochondrial import. The presented data clearly shows that the method works and is widely applicable in the field of mitochondrial biology. With some additional required controls and validations we would therefore find it suitable for publication in *eLife*. Below please find the requested additional experiments and textual changes:

Required changes

1) Results, Figure 2E and text: The FACS experiment (which is actually not FACS but flow cytometry because there is no cell sorting included) requires an additional control of a strain transformed with an empty vector (EV) to show whether the Pgk-beta11 has the same intensity as the EV control or higher.

2) Figure 2B: It is critical to compare the Pgk1 strain with the EV control in terms of fluorescence intensity to see if there is really no background. Please display these two micrographs in the GFP channel with the contrast enhanced in the same way so that the background signals are clearly visible and readily comparable. Other micrographs in this panel can be displayed with the same contrast too, if not oversaturated. Alternatively, fluorescence signal quantification can be added, so that the signals can be compared to the control experiment.

3) To support the conclusion from your method that certain amino acyl tRNA synthetases are dually localized to the cytosol and to the mitochondrial matrix we request some validation of this data by an additional method such as biochemical fractionation or functional data for the relevance of these tRNA-synthetases for mitochondrial protein synthesis

4) Cloning of fragments of proteins is dangerous. It was shown already in 1987 by Ed Hurt and Jeff Schatz (Nature 325, 599-503) that the subcloning of fragments of cytosolic proteins causes their artificial and misleading import into mitochondria. Thus, it is essential that the C-terminal reporter, at least of key experiments, is verified by being fused by use of expression cassettes that are integrated into the genome. This also prevents artifacts from overexpression.

5) The claim that the current approach is superior to other split gene approaches in which both fragments of the split protein are translated in the cytosol should be further validated. At present it is based on one experiment in which a sub-optimal MTS was used for the nuclear encoded fragment (GFPβ1-10 at the C terminus of a the full GatF protein of 183aa which is claimed to have a strong MTS but this is not shown). The strongest used MTS in yeast is the 69 most N terminal amino acids of Su9. Using this MTS it has been shown that precursors are difficult to detect even in pulse chase experiments and there are no precursors accumulating in the cytosol unless the cells are under severe stress. Hence the authors should either prove that GatF MTS is an optimal signal, or else redo the control experiments with the Su9 mTS or else tone down their statement.

6) As an extension to point 5 – the sensitivity of the current method is not clear. If this new split system has a very low expression of GFPβ1-10 from the mitochondrial genome, it may not be sensitive enough to identify novel low expressed proteins in mitochondria. We would like to see an evaluation of how sensitive the GFPβ1-10 expressed from the mitochondrial genome is in detecting low-abundance mitochondrially targeted counterparts attached to a GFPβ11ch. Optimally this would be compared in sensitivity (and not only accuracy) to the nuclear expressed MTS-GFPβ1-10.

7) The authors use either Pam16β11ch or Atp4β11ch as their positive mitochondrial controls. but both are membrane proteins. Please also use one soluble protein that is less abundant as a control.

8) The authors use as a negative control, a GFPβ11ch tagged version of Pgk1, which they claim is a commonly used cytosolic marker. However, Pgk1 is annotated as having a mitochondrial pool (see for example SGD) and this may explain the background that can be seen. Maybe a purely cytosolic protein would be a better control?

---

## [Author Response]

Required changes1) Results, Figure 2E and text: The FACS experiment (which is actually not FACS but flow cytometry because there is no cell sorting included) requires an additional control of a strain transformed with an empty vector (EV) to show whether the Pgk-beta11 has the same intensity as the EV control or higher.

We agree, this is a control missing in our experiment. We will measure the fluorescence intensity by flow cytometry of the BiG Mito-Split-GFP strain bearing the empty vector used for GFP_β11ch_ tagging and compare it to the BiG Mito-Split-GFP strain expressing Pgk1_β11ch_ and Pam16 _β11ch_ to stay in similar conditions. It is now indicated in the revised manuscript that additional experiments are needed to verify whether the BiG Mito-Split-GFP system can be used for systematic screening.

2) Figure 2B: It is critical to compare the Pgk1 strain with the EV control in terms of fluorescence intensity to see if there is really no background. Please display these two micrographs in the GFP channel with the contrast enhanced in the same way so that the background signals are clearly visible and readily comparable. Other micrographs in this panel can be displayed with the same contrast too, if not oversaturated. Alternatively, fluorescence signal quantification can be added, so that the signals can be compared to the control experiment.

As requested, we have enhanced the contrast in this figure and also added, in the source data file, new micrographs of the BiG Mito-Split-GFP strain expressing Pgk1_β11ch_, with the same “enhanced” and “not enhanced” contrast settings. In addition, we have also compared the fluorescence intensity of GFP and MitoTracker Red CMXRos signals using ImageJ software across cells of the BiG Mito-Split-GFP cells transformed with the positive controls (Pam16_β11ch_, Atp4_β11ch_), the cERS and the negative controls EV and Pgk1_β11ch_. This analysis, that shows perfect colocalization of Pam16_β11ch_, Atp4_β11ch_ and cERS_β11ch_ but not of the EV or Pgk1_β11ch_, was added as a new panel in Figure 2—figure supplement 1A which is mentioned in the revised manuscript (subsection “The BiG Mito-Split-GFP system restricts fluorescence emission to mitochondrially-localized proteins”). We will also add micrographs of the BiG Mito-Split-GFP strain expressing His3_β11ch_ that will be taken with the same enhanced contrast settings. His3_β11ch_ is another cytosolic control that we had already generated before COVID-19 confinement and that will be provided as indicated in the revised manuscript.

3) To support the conclusion from your method that certain amino acyl tRNA synthetases are dually localized to the cytosol and to the mitochondrial matrix we request some validation of this data by an additional method such as biochemical fractionation or functional data for the relevance of these tRNA-synthetases for mitochondrial protein synthesis

We are conscious that the data on _cyte_FRS2 and _cyte_HRS are intriguing because the most recently published mitoproteomes based on purification of mitochondria followed by mass spectrometry identification (Vogtle et al., 2017) did not detect _cyte_FRS2 and _cyte_HRS in mitochondrial extracts. But this is not the only discrepancy that can be found between previous work and ours concerning caaRSs that can potentially relocate to mitochondria. For example, previous studies (Rinehart et al.,2005) and recent mitoproteomes (Vogtle et al., 2017) suggest that cQRS might be imported inside mitochondria. However, we unambiguously demonstrated both genetically and functionally that cQRS is not imported inside mitochondria (Frechin et al., 2009); and the micrographs of the BiG Mito-Split-GFP strain expressing cQRS_β11_ unquestionably confirmed these previous results (Figure 3). Likewise, we (Frechin et al., Genes & Dev. 2009, Frechin et al., 2014) and also others (Vogtle et al., 2017) repetitively proved the presence of cERS_β11ch_ in mitochondria, a result confirmed by micrographs the BiG Mito-Split-GFP strain expressing cERS_β11_ (Figure 2 and 3), which corresponds exactly to the verification asked by the reviewers. Moreover, the GFP signals for _cyte_HRS_β11ch_ and _cyte_FRS2_β11ch_ (Figure 3) seems to be even significantly stronger than for cERS_β11ch_ while being expressed at similar levels (Figure 3—figure supplement 1), suggesting that _cyte_HRS_β11ch_ and _cyte_FRS2_β11ch_ might even be more efficiently imported inside mitochondria than cERS_β11ch_. We therefore do not see how immunoblotting extracts of purified mitochondria would enhance the reliability of our approach especially considering that the idea behind the BiG Mito-Split-GFP is rightly to avoid biochemical fractionation and to show that alternative ways do exist to clearly identify mitochondrial proteins and echoform by simple microscopy.

The alternative request to provide data showing that the mitochondrial echoforms of _cyte_FRS2 and _cyte_HRS might participate to mitochondrial translation seems to us very hazardous because relocating caaRSs usually exert non-translational functions in the new compartment they reach (Yakobov et al., 2017). We are therefore inclined to believe that the mitochondrial echoforms of _cyte_FRS2 and _cyte_HRS will very probably not participate to mitochondrial translation. Furthermore, as these forms are essential for cytosolic translation, generating mutants of _cyte_FRS2 and _cyte_HRS that have conserved their function in cytosolic translation while being impaired for their mitochondrial role, whatever this role might be, is far from being an obvious, fast and effortless task. It will inevitably require that we first identify the cryptic MTS of both _cyte_FRS2 and _cyte_HRS and second that their removal doesn’t impair the cytosolic activity of both enzymes. If this is the case, then we might be able to decipher the mitochondrial roles of _cyte_FRS2 and _cyte_HRS. However, I hope that the reviewers can understand that this will not be a swift verification but will rather constitute, by itself, a whole new research project.

However, if having another validation of the mitochondrial import of _cyte_FRS2 and _cyte_HRS appears to be crucial to the reviewers, we will check by immunoblotting of pure mitochondrial extracts the presence of these echoforms and add it to the BioRxiv file that will be linked to our manuscript.

4) Cloning of fragments of proteins is dangerous. It was shown already in 1987 by Ed Hurt and Jeff Schatz (Nature 325, 599-503) that the subcloning of fragments of cytosolic proteins causes their artificial and misleading import into mitochondria. Thus, it is essential that the C-terminal reporter, at least of key experiments, is verified by being fused by use of expression cassettes that are integrated into the genome. This also prevents artifacts from overexpression.

We assume that the reviewers refer to the experiments shown on Figure 4A and 4C in which we show the micrographs we obtained by fusing to GFP_β11ch_, various fragments of the N-terminal GTS-like domain of cERS. We do agree with the reviewer that mislocalization can be triggered with N-terminal protein fragments and lead to false positive identification of organellar-targeted proteins, and that verifying the localization of the corresponding C-terminal part, is a needed control. We believe that the data shown in Figure 4A (now Figure 4B of the revised manuscript) provide these verifications. Indeed, they show that the 200 N-terminal residues of cERS (cERS_β11ch_-N3) trigger mitochondrial targeting of GFP_β11ch_ while, as expected, removing them from cERS_β11ch_ (cERS_β11ch_-∆N2) prevent its mitochondrial import. Likewise, the 30 fist N-terminal residues of cERS (cERS_β11ch_-N1) trigger mitochondrial targeting of GFP_β11ch_ while removing them from cERS_β11ch_ cERS_β11ch_-∆N1) prevent its mitochondrial import. As was done by Hurt and Schatz in their 1987 Nature paper, we generated the N-terminal fragments of cERS according to the structure that was published by Simader and coworkers in their 2006 NAR paper, making sure that the truncations were exclusively done in loops and not in the middle of an a-helix or a β-strand.

Moreover, the DHFR fragment that was identified by Hurts and Schatz, in their 1987 Nature paper and which is acting as an artificial MTS, contained an a-helix which resembles a mitochondrial pre-sequence: 3 positively-charged residues (interspaced by 3 aa residues), 1 Glu and 3 Thr residues. To the contrary, the 30 first aa residues of cERS contain one a-helix of 10 residues that only contains 1 Arg but also 1 Glu (see Figure 4B of the revised manuscript) which, by far, does not correspond to a *bona fide* mitochondrial pre-sequence that could trigger artificial mislocalization of a fused peptide, unless it is really a new type of MTS whose mechanistic traits have yet to be deciphered.

If we well understood, the second concern of the reviewers were artificial localization originating from overexpression. We would like to emphasize that caaRSs are naturally abundant proteins in yeast cells (30.000-40.000 exemplar / cell on average with around 10 hours half-life – SGD) and the cERS fragments and truncated variants were expressed under the dependence of a GPD promoter which has a strength similar to that of the aaRSs endogenous promoters and comparable to the ADH promoter that was used by Hurt and Schatz in their 1987 Nature paper. Interestingly, in their paper, Hurts and Schatz unambiguously show that despite comparable overexpression of various parts of the 85 N-terminal aa residues of DHFR, only the one containing the cryptic MTS (see previous paragraph) triggers mitochondrial import in vivo. This shows that overexpression is less prone to trigger mitochondrial mistargeting than the artificial presence of a mitochondrial pre-sequence.

It is true that we did not chromosomally integrate the GFP_β11ch_-tagged N-terminal fragments of cERS_β11_ or the N-terminally truncated cERS_β11ch_ variants but we expressed them from a low-copy plasmid that mimics the number of chromosomal gene copies. In addition, the micrographs of the BiG Mito-Split-GFP strain in which we chromosomally integrated cERS_β11ch_ expressed under the dependence of its natural promoter are comparable to the micrographs obtained with the BiG Mito-Split-GFP strain transformed with a single-copy plasmid expressing cERS_β11ch_ under the dependence of the GPD promoter. Therefore, we don’t think that the mitochondrial localizations we observe in Figure 4A (now Figure 4B of the revised manuscript) are artificially caused by the truncations we made or overexpression.

However, to overcome problems originating from the nomenclature we used in Figure 4A (now Figure 4B of the revised manuscript), we changed it and to simplify this figure we have added a new figure (Figure 4A of the revised manuscript) showing the drawings of the N-terminal fragments and truncated variants of cERS. We also added the aa sequence of thee a-helices on the schematized secondary structure of the N-terminal par of cERS.

Concerning the N-terminal part of cCRS, as we mentioned in the revised manuscript, we were unable to get *E. coli* transformants bearing a plasmid with the full-length cCRS gene, despite repeated attempts and using different cloning procedures (Gateway-, Gibson-, regular restriction enzyme-mediated cloning procedures). However, while we were doing our experiments with the N-terminal domain of cCRS, the 2019 paper by Nishimura and coworkers (J. Biol. Chem.294 13781-13788) proved that the mitochondrial and cytosolic echoforms are generated through a combination of alternative transcription and translation initiation. Given that the N-terminal sequence of cCRS we used corresponds to that present in the mitochondrial echoform characterized by Nishimura and coworkers, we did not further pursue on trying to get the C-terminal part of cCRS cloned because our BiG Mito-Split-GFP data were in agreement with theirs.

5) The claim that the current approach is superior to other split gene approaches in which both fragments of the split protein are translated in the cytosol should be further validated. At present it is based on one experiment in which a sub-optimal MTS was used for the nuclear encoded fragment (GFPβ1-10 at the C terminus of a the full GatF protein of 183aa which is claimed to have a strong MTS but this is not shown). The strongest used MTS in yeast is the 69 most N terminal amino acids of Su9. Using this MTS it has been shown that precursors are difficult to detect even in pulse chase experiments and there are no precursors accumulating in the cytosol unless the cells are under severe stress. Hence the authors should either prove that GatF MTS is an optimal signal, or else redo the control experiments with the Su9 mTS or else tone down their statement.

We did not there *per se* claim that our approach is superior but rather that it is significantly more reliable in avoiding false positives than when both fragments are translated in the same compartment. However, the reviewers are right, we did not present in the manuscript, data to support that GatF’s MTS is an optimal MTS. But, taking into account all currently existing data on GatF in the literature, I do not see how the reviewers came to the conclusion that GatF’s MTS is sub-optimal, either.

We generated the MTS-based Split-GFP fragments to be able to evaluate the possibility that the cytosolically-translated MTS-GFP_β1-10_ could bind Protein X-GFP_β11_ before its import and thus mislabel mitochondria by assembling at the surface of mitochondria rather than inside. This is also why we used Pgk1_β11ch_ as our cytosolic control (see reviewers’ comment #8) because a fair proportion of this cytosolic protein was demonstrated to be located at the external surface of mitochondria. We were aware that the strength of the MTS we would fuse to GFP_β1-10_ had to be taken into consideration because its strength might impact the time during which the MTS-GFP_β1-10_ resides at the surface of mitochondria. We assumed that the weaker the MTS would be, the longer the MTS-GFP_β1-10_ would accumulate near the mitochondrial surface rather than being imported inside the organelle. This is why we generated two MTS-GFP_β1-10_ and compared their efficiency in generating a mitochondria-specific GFP labeling that could be detected by epifluorescence microscopy, when co-expressed with the dual-localized cERS_β11ch_. The two MTS we compared were the entire GatF protein that we had both functionally and structurally characterized, and the MTS of the mitochondrial malate dehydrogenase (Mdh1) (see Author response image 1).

**Author response image 1. sa2fig1:** Efficiency of mitochondria-specific GFP labeling induced by MTS-MDHβ1-10 compared to that of GatFβ1-10. *Saccharomyces cerevisiae* gus1∆ strain complemented with the pRS414 plasmid expressing cERSβ11ch was transformed with a pRSX expressing either MTS-MDHβ1-10 (A and C) or GatFβ1-10 (B and D) and grown either on SC-Glu (A and B) or SC-Gly (C and D). Epifluorescence micrographs were taken with an AXIO Observer d1 (Carl Zeiss) epifluorescence microscope using a 100 × plan apochromatic objective (Carl Zeiss) and processed with the Image J software. Arrowheads: mitochondria.

As can be seen in B and D, GatF_β1-10_ allows efficient and specific labeling of mitochondria with almost no cytosolic background both in SC-Glucose and SC-Glycerol media. Conversely, in SC-Glucose, MTS-MDH_β1-10_ (A) does not yield a mitochondrial GFP signal that can be distinguished from the residual cytosolic one. In SC-Glycerol, one can start to distinguish mitochondria when MTS-MDH_β1-10_ is used (C), but the mitochondrial GFP signal is weak and there is still a significant cytosolic GFP signal. Because GatF_β1-10_ generates a strong mitochondrial GFP signal with almost no contaminated cytosolic GFP emission, compared to MTS-MDH_β1-10_, we concluded that GatF can be considered as a strong MTS. We did not evaluate GatF’s import strength compared to that of the MTS of the heterologous *Neurospora crassa* Atp9 subunit, but we hope that our comparative study using endogenous *S. cerevisiae* MTSs will convince the reviewers that, *a minima*, GatF can be considered as an efficient MTS; and thus supports our conclusion that the BiG Mito-Split-GFP constitutes indeed a significantly more reliable approach for visualizing mitochondrial echoforms of dual-localized proteins. We nevertheless completely modified this part of the manuscript which is now: “We next evaluated whether the BiG Mito-Split-GFP approach represents a significant technical advance compared to the existing MTS-based Split-GFP methods that are currently used. To this end, we constructed cells (with a wild type mitochondrial genome) that co-express in the cytosol the mitochondrial protein GatF (with its own MTS) fused at its C-terminus with GFP_β1-10_ (mtGatF_β1-10_) and either cERS_β11ch_ (dual localized, positive control) or Pgk1_β11ch_ (cytosolic, negative control) (Figure 2F, left panel). As expected, a strong and specific mitochondrial fluorescent signal was obtained with cERS_β11ch_ (Figure 2F, right panel).”

6) As an extension to point 5 – the sensitivity of the current method is not clear. If this new split system has a very low expression of GFPβ1-10 from the mitochondrial genome, it may not be sensitive enough to identify novel low expressed proteins in mitochondria. We would like to see an evaluation of how sensitive the GFPβ1-10 expressed from the mitochondrial genome is in detecting low-abundance mitochondrially targeted counterparts attached to a GFPβ11ch. Optimally this would be compared in sensitivity (and not only accuracy) to the nuclear expressed MTS-GFPβ1-10.

We agree that knowing how sensitive the BiG Mito-Split-GFP method is, is an important issue, which we did not sufficiently address. The yeast proteins we’ve tested are indeed far from being weakly expressed: Pam16 (3000 copies), Atp4 (30.000-40;000 copies), caaRSs (13.000-70.000 copies). By looking at most recent proteomics data reporting on the copy number of proteins found inside mitochondria (Vogtle et al., 2017), we noticed that mitochondrial GatF protein (*GTF1*) is a low-expressed protein (40-80 copies in cells grown in YPGal or YPGly). Since we already generated a plasmid expressing GatF_β11ch_ under the dependence of a GPD promoter, we propose to swap this promoter by the *GTF1* endogenous one and to compare the intensity the BiG Mito-Split-GFP strain expressing GatF_β11ch_ under the dependence of its own promoter to that of cERS_β11ch_ for example. This additional experiment to be reported later and its importance, is mentioned in the revised manuscript.

7) The authors use either Pam16β11ch or Atp4β11ch as their positive mitochondrial controls. but both are membrane proteins. Please also use one soluble protein that is less abundant as a control.

We do not fully agree with the reviewers on this point, at least as far as Pam16 is concerned. This is not a *bona fide* inner membrane protein. It has been shown it is translocated into the mitochondrial matrix where it associates to the mtHsp70 before reaching the TIM complex where it interacts with Tim44 (a true integral membrane protein). This seem to be a common feature of matrix proteins because when we looked at the most recent data that report localization of mitochondrial proteins (Vogtle et al., 2017, supplementary file 4), the degree of precision that authors can reach for the sub-mitochondrial does not allow the separation between the matrix and the inner mitochondrial membrane. In this report, authors separate submitochondrial compartments into: OM (outer membrane), IMS/IM (intermembrane space / Inner Membrane) and matrix/IM (matrix/Inner Membrane). We are wondering if one can find a matrix-restricted protein which is not a peripheral mitochondrial inner membrane protein because it, at least, transiently can interact with proteins or protein complexes that are embedded in the inner membrane.

We, therefore, do not think that another so-called matrix protein – probably annotated as a matrix/inner membrane protein – would be a better positive control than Pam16, unless the reviewers have a particular matrix-restricted protein in mind that we haven’t come across in our analysis of the literature.

8) The authors use as a negative control, a GFPβ11ch tagged version of Pgk1, which they claim is a commonly used cytosolic marker. However, Pgk1 is annotated as having a mitochondrial pool (see for example SGD) and this may explain the background that can be seen. Maybe a purely cytosolic protein would be a better control?

If we may, our claim that Pgk1 is a cytosolic marker commonly used rests on the large number of studies in which Pgk1 has indeed been used as a cytosolic marker on WB. As we mentioned in our initial manuscript, Pgk1 localizes in part at the surface of mitochondria but once purified mitochondria are treated with proteinase K, this pool totally disappears, showing that Pgk1 is not internalized inside mitochondria (Cobine et al., 2004; Levchenko et al., 2016; Kritsiligkou et al., 2017). We changed this paragraph in the revised manuscript. Moreover, as already mentioned in our answer to the 2^nd^ major comment, we will provide micrographs of the BiG Mito-Split-GFP strain expressing His3_β11ch_ as another negative control in the BioRxiv addendum that will be linked to our revised manuscript. This is now mentioned in of the revised manuscript.